# Double and single stranded detection of 5-methylcytosine and 5-hydroxymethylcytosine with nanopore sequencing

Dominic Oliver Halliwell [1] ✉, Floris Honig [1], Stefan Bagby [1], Sandipan Roy [2] & Adele Murrell [1] ✉

5-methylcytosine (5mC) and 5-hydroxymethylcytosine (5hmC) are modified versions of cytosine in DNA with roles in regulating gene expression. Using whole genomic DNA from mouse cerebellum, we benchmark 5mC and 5hmC detection by Oxford Nanopore Technologies sequencing against other standard techniques. In addition, we assess the ability of duplex base-calling to study strand asymmetric modification. Nanopore detection of 5mC and 5hmC is accurate relative to compared techniques and opens means of studying these modifications. Strand asymmetric modification is widespread across the genome but reduced at imprinting control regions and CTCF binding sites in mouse cerebellum. Here we demonstrate the unique ability of nanopore sequencing to improve the resolution and detail of cytosine modification mapping.

Chemically modified versions of cytosine, 5-methylcytosine (5mC) and 5-hydroxymethylcytosine (5hmC), affect transcriptional regulation and play important roles in many biological processes and diseases, including development, ageing, and cancer[1]. In eukaryotic genomes, these modifications are enriched at motifs known as CpG sites, dinucleotide positions in which cytosine is followed immediately by guanine. DNA methyltransferase (DNMT1) maintains methylation at these sites symmetrically, at the cytosine base on both the forward and reverse strand. Asymmetrical modification is largely a feature of cell division, with the production of a temporarily unmethylated template that is then methylated by DNMT1[2]; however, recent reports suggest that hemi-methylation, where 5mC is present on only one strand, can be stably maintained after cell division[3], and can affect DNA-transcription factor interactions[4].

Oxidative conversion of 5mC by the Ten-Eleven Translocation (TET) family of proteins produces 5hmC[5]. 5hmC is a stable modification important for maintaining pluripotency in embryonic stem cells, neural development, and tumorigenesis[6,7], but can also be removed through both passive and active enzymatic demethylation.

Genomic imprinting is an epigenetic phenomenon where some mammalian genes silence one parental allele during gametic development[8]. A collection of stable allele-specific differentially methylated regions (DMRs), located at regulatory elements that act as imprinting control centres, offers an excellent model system for studying DNA methylation dynamics in mammalian cells. Allele-specific transcription is regulated by

methylation-sensitive transcription factors that bind within the DMRs. A notable example being CTCF, which structures allele-specific 3D chromatin conformation at selected imprinted loci[9], as well as genome-wide interactions between distant regulatory elements[10].

Most DNA methylation detection assays are performed using bisulphite sequencing techniques[11]. Sodium bisulphite causes cytosine bases to deaminate, converting them to uracil, which is then replaced by thymine during PCR. 5mC and 5hmC resist deamination and amplify as cytosine. Sequencing the product provides a base-resolution binary readout of converted (unmodified) and non-converted (modified) bases. The process damages DNA, however, and destroys up to 90% of the initial DNA template[12]. Subsequent PCR steps introduce PCR bias and generate products with reduced sequence complexity, leading to low mapping rates and uneven genome coverage[13,14]. Since both 5mC and 5hmC are protected from bisulphite-mediated deamination, different treatments are also required to distinguish between 5mC and 5hmC.

Oxidative bisulphite sequencing (oxBS-seq)[15] and TET-assisted bisulphite sequencing (TAB-seq)[16] were developed as means of sequencing 5hmC at base-pair resolution. These methods involve protection of either 5mC (oxBS-seq) or 5hmC (TAB-seq) from bisulphite-mediated deamination, leaving a binary readout of the protected modification only. The 5mC-specific signal from oxBS-seq can also be subtracted from the standard bisulphite reference to predict the presence of 5hmC; however, sample variation can lead to logical inconsistencies, such as the calculation of

[1]Department of Life Sciences, University of Bath, Bath, UK. [2]Department of Mathematical Sciences, University of Bath, Bath, UK. ✉e-mail: doh28@bath.ac.uk; amm95@bath.ac.uk

negative hydroxymethylation[17]. Inefficient chemical conversion is the primary sources of read-level error within these methods. Within TAB-seq, protection efficiency for 5hmC can be as low as 92%[16], resulting in its deamination and false negative reading; conversely, 5mC-to-T conversion can be as low as 96%[18], leading to false positive detection of methylated bases as 5hmC. Given that 5mC is, in tissues like mouse embryonic stem-cells, 14-fold more abundant than 5hmC[5], it is possible that many TAB-seq-derived 5hmC detections represent this error. Where 5mC requires relatively low input depth to be sequenced reliably, at roughly 5-10x in whole-genome bisulphite sequencing[13], deep sequencing is essential to confidently predict 5hmC, with the required depth increasing when the rate of incomplete conversion is higher[16], or when 5hmC abundance in a tissue is lower[15]. Several methods have been developed to reduce these limitations, including TAPS sequencing and related methods[19,20], and ACE-seq[21]. Although accurate and less deleterious to sample DNA, these methods still rely on multiple chemical or enzymatic steps and remain vulnerable to PCR-induced biases.

Immunoprecipitation-based approaches are used for genome scale 5mC and 5hmC sequencing. Methyl- and hydroxymethyl-DNA immunoprecipitation (MeDIP/hMeDIP) employ an anti-5mC/5hmC antibody to isolate the modified DNA[22,23], which is then amplified and sequenced, producing "peaks" of coverage over regions where modifications are enriched identical to chromatin (ChIP) sequencing[24]. As whole fragments are pulled down, these methods lack base resolution. It is also difficult to distinguish instances of nonspecific binding of antibody to unmodified DNA, which can produce false positive enrichment peaks accounting for a large proportion of peak regions[25].

Third generation sequencing technologies produce a direct readout of canonical bases and of some modified bases without chemical conversion or PCR amplification. Two such technologies, PacBio's Single Molecule, Real-Time (SMRT) sequencing[26] and Oxford Nanopore Technologies (ONT) sequencing[27], support detection of different base modifications, including DNA methylation. SMRT sequencing detects polymerase-catalysed addition of nucleotides to a DNA template as fluorescent pulses; polymerase kinetics are affected by modified bases, which allows the type of base to be inferred[28].

In nanopore sequencing, blockage differences in ion channel current are recorded to detect and identify the base sequence present in a nanopore at any moment. Modifications such as 5mC and 5hmC, and RNA modifications such as 6-methyladenine (m6A)[29], induce detectable differences in blockage signal relative to the canonical base[30]. These differences are decoded using software such as nanopolish[31] and signalAlign[32], which are both built on hidden Markov models, or DeepSignal[33], and Megalodon[34], which both use artificial neural networks. Predictions from several tools, including nanopolish, DeepSignal, DeepMod[35], and Oxford Nanopore Technologies' Guppy, Megalodon, and Tombo, are also included in METEORE, a consensus approach capable of improving accuracy over individual tools[36]. 5mC detection by these tools correlates well at base level with bisulphite data ($r > 0.8 - 0.95$)[36,37], even at lower coverage depths ($d \geq 3$)[38]. 5hmC-trained base-calling models were largely lacking, however, until the development of Remora[39].

The latest generation of Oxford Nanopore Technologies' flow cells (R10.4.1HD) offers further capabilities suited to epigenetics research. In standard "simplex" sequencing, double-stranded input DNA is unwound and one strand passes through a nanopore while the other is discarded or sequenced in a different pore. R10.4.1HD flowcells improve second strand capture rate, enabling the two strands of a dsDNA molecule to proceed successively through a nanopore. The two reads are then paired informatically. This technique, "duplex" sequencing, improves base-calling and mapping accuracy via consensus from both strands of a DNA molecule. Via simultaneous examination of the modification status of both strands, moreover, duplex sequencing permits determination of strand modification symmetry and asymmetry.

In this study we have used Oxford Nanopore Technologies' PromethION to generate whole genome sequences of mouse cerebellum for benchmarking, comparing more than 1.6 billion base-calls encompassing

12.3 million CpG dinucleotide positions with public oxBS and TAB-seq data. In addition, we verified the sensitivity of the base-caller to higher input 5hmC via direct sequencing with a MinION of a 5hmC-enriched hMeDIP library. Finally, we used nanopore duplex sequencing to query locus-level modification symmetry at imprinted loci and CTCF binding sites.

## Results

### Accuracy of raw read detection of 5mC from methylation standards

We sequenced two commercially available human DNA methylation standards (Zymo, D5013), a whole genome amplification (WGA)-produced modification-negative control and an enzymatically methylated 5mC-positive control. The modification negative control has no possibility of base modification; thus, all modified base detections represent false positives. The methylation positive control, which is enzymatically methylated after WGA, has a high degree of methylation (>95%) as reported by the manufacturer. All 5hmC detections represent classification errors. Raw read accuracy, which is the accuracy of modified base detection at a single base in one read, was very high for 5mC, with a precision of 0.99 and recall of 0.97 (Fig. 1a; Table 1).

Previous reports highlight a vulnerability of nanopore sequencing to mapping mismatches at regions of especially high GC content[40]. To analyse its effect on modification detection, we inspected rates of false positive modification detection (FPR) as a function of local GC content. This showed a strong correlation between GC content and FPR ($r = 0.76$; $p \geq 0.001$) (Fig. 1b), with GC contents substantially higher than the genomic mean (41.06%) experiencing the highest error rates.

Although we could not find any specific sequence motif in the 12-mer sequence up/downstream of a false positive modified base-call, there was a higher proportion of G or C base detections than expected from the genomic mean for both false positives of 5mC ($GC = 0.56$) (Fig. 1c) or 5hmC ($GC = 0.59$)(Fig. 1d).

Certain genomic elements may be predisposed to false positive error due to GC content; namely, Alu repeats (mean 51.3% GC)[41], satellite repeats (mean 47.1% GC), and low complexity regions or simple repeat sequences (GC variable), such as tandem repeats. GC-rich CpG islands (CGI) (mean 68.6% GC), which are important for transcriptional regulation, could also be vulnerable to this effect. Indeed, in the modification-negative standard, FPR above the genomic mean was noted in low complexity repetitive sequences (Fig. 1e). By contrast, in the methylation-positive standard (Fig. 1f), false 5hmC detection was largely consistent with the genomic mean 5hmC FPR for most contexts except for simple repeat elements, where 5hmC FPR appears to double.

False positive detection of 5hmC was present in both the unmodified and methylated standards. There was an almost five-fold difference in false 5hmC detection between the unmethylated (0.0024) and methylated (0.011) standards (Fig. 1g). This indicates a tendency for the base-caller to mistake true methylated positions for hydroxymethylation.

### Rates of calling of modified bases: nanopore is highly consistent with bisulphite-based sequencing methods

To assess modified base detection from ex vivo tissues and compare rates of base modification with orthologous techniques, cerebellar tissues from two 8-week-old female mice were sequenced in duplicate, producing four whole genome datasets (median genomic depth 29-32x) (Table S1). CpG modification calls were extracted from this, with median CpG depths of 15–17x (stranded) (Fig. S1a). Samples were selected to match a publicly available archive of sequence data, containing whole genome oxidative bisulphite sequencing (oxBS-seq) for two biological replicates and TET-assisted bisulphite sequencing (TAB-seq) for three biological replicates[42]. The datasets produced using these techniques were lower coverage depth, with a median CpG depth of 3x (stranded) for TAB-seq (Figs. S1b), and 2-6x for oxBS-seq (Fig. S1c). To account for higher rates of FPR found in some repeat contexts, repeats were soft-masked.

Higher genomic depth in the nanopore sequence data (Fig. S1d) provides a larger sample of sequence reads per cytosine (Fig. S1e). Previous studies in

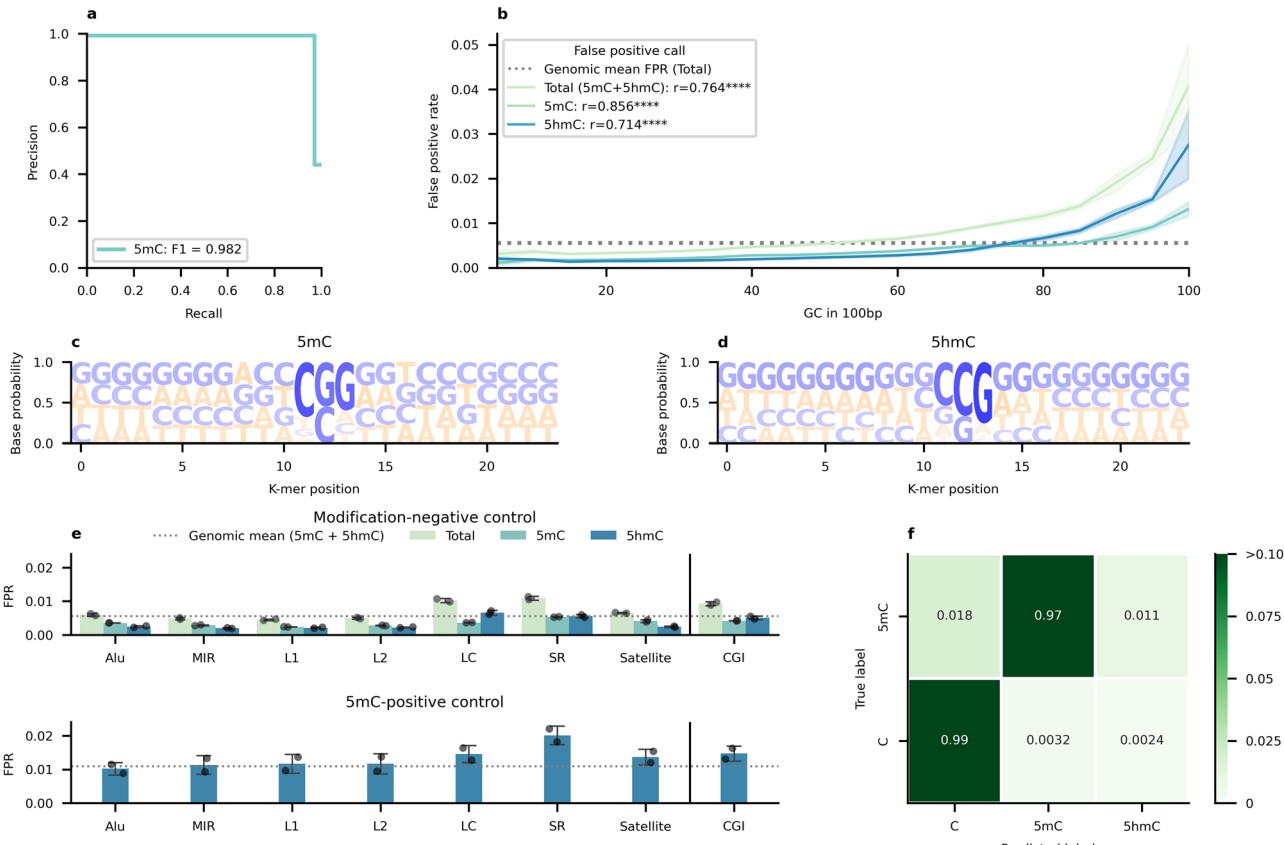

**Fig. 1 | Raw read accuracy of nanopore modified base detection using human whole genomic controls. a** Precision-Recall curve for 5mC detection using unmodified ($N = 2$) and methylated controls ($N = 2$). 5hmC detections are included as error, due to the absence of 5hmC in either control sample. Base-calls from both replicates of each control are counted and down sampled to 100,000,000 base-calls. **b** False positive rate of modified base detection from the unmodified control as a function of local GC content. CpG base-calls are binned into non-overlapping 100 bp windows and GC percentage calculated using the mm39 (GRCm39) reference genome. Bands indicate a 95% confidence interval across replicates ($N = 2$). Logo representation of the 12-mer sequence up/down stream of false positive modified base detections for 5mC (**c**) and 5hmC (**d**). Higher base probabilities are shaded and stacked top-down. Includes reads from both unmodified DNA standards ($N = 2$). **e** Rates of false positive modified base detection across classes of repetitive, low-complexity, or CpG island sequences, including (top) 5mC and 5hmC false positives in the modification-negative control ($N = 2$), and (bottom) 5hmC false positives in the 5mC-positive control ($N = 2$). LC: Low Complexity. SR: Simple Repeat. Error bars indicate 1 s.d. **f** Confusion matrix for predicted and ground-truth base-calls. Base-calls across all replicates are counted, considering C to be the ground-truth state of all base-calls in the unmodified controls ($N = 2$), and 5mC in the methylated controls ($N = 2$). ****$p < 0.0001$.

## Table 1 | 5mC base detection statistics using methylation standards

| True Positive Rate (TPR) | False Negative Rate (1-TPR) | False Positive Rate (FPR) | Precision $\left(\frac{TPR}{TPR+FPR}\right)$ | Recall $\left(\frac{TPR}{TPR+FNR}\right)$ | F1-score $\left(2\left(\frac{Precision \cdot Recall}{Precision + Recall}\right)\right)$ |
|---|---|---|---|---|---|
| 0.97 | 0.03 | 0.0056 | 0.99 | 0.97 | 0.98 |

Base-calls from both the unmodified negative standard and 5mC-positive standard are concatenated and, for computational purposes, down-sampled to 100,000,000 base-calls.

bisulphite based libraries have highlighted a negative correlation between GC content and sequencing depth[43]. In our study we noted significant negative correlation with depth (Fig. S1f) and local GC content for all methods (Nanopore: $\rho = -0.53, p \leq 0.001$; oxBS-seq: $\rho = -0.71, p \leq 0.001$; TAB-seq: $\rho = -0.79, p \leq 0.001$), with oxBS-seq and TAB-seq showing the most prominent loss of coverage even below the genomic mean GC content (41.7%) (Nanopore: $\rho = -0.54, p \leq 0.001$; oxBS-seq: $\rho = -0.71, p \leq 0.001$; TAB-seq: $\rho = -0.59, p \leq 0.001$). Thus GC-related effects may explain the lower depth of coverage found in promoters and CpG islands (Fig. S1h), as well as general loss of coverage (Fig. S1i).

Modified base detection is similar in the nanopore sequencing and bisulphite-based data. CpG methylation in either method follows a bimodal distribution (Fig. 2a), with two maxima of CpG positions that are either completely methylated or completely unmethylated. Fewer 5mC detections are made by the nanopore base-caller (60.3–62.8%) than by oxBS-seq (66.3–68.8%) (Table S2); however, the difference is non-

significant (Welch's T-Test: $p = 0.08$; Cohen's $d$: $d = 4.06$), potentially on account of sample size differences. False negative 5mC detection is unlikely to fully account for this, with the difference between means (6.0%) larger than that expected from the false negative error (3.0%) noted above, potentially indicating a bias favouring the detection of methylated bases in oxBS-seq.

5hmC is unimodally distributed around 0% in either method, with a maximum reflecting a high density of CpG sites in which 5hmC is not found (Fig. 2b). As above, the proportion of base-calls detected as 5hmC is also lower in the nanopore data, ranging from 9.6 to10.4%, and 11.0–11.8% in TAB-seq (Welch's T-Test; $p = 0.004$; Cohen's $d$: $d = 4.10$). In the absence of a ground-truth 5hmC control, this difference (difference between means: 1.6%) may approximate the rate of false negative error for 5hmC detection by nanopore sequencing. 5hmC was significantly more likely to be detected at any given CpG position at least once, likely due to the difference in depth (Welch's T-Test; $p = 0.01$; Cohen's $d$: $d = 3.21$).

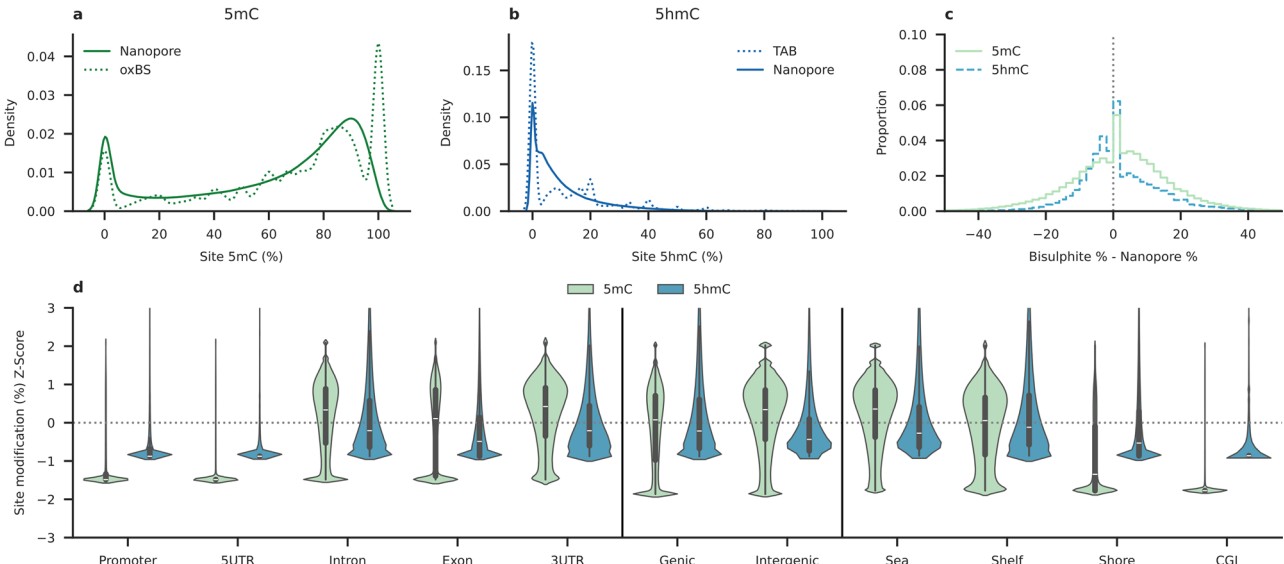

**Fig. 2 | CpG-resolution modified base detection using nanopore sequencing and bisulphite methods.** Base-calls from matched CpG positions are concatenated across replicates in all plots. Density distributions of (**a**) methylation, or (**b**) hydroxymethylation, as the percentage modified reads at individual CpG sites, as found by nanopore sequencing (solid line) and the orthologous bisulphite method (dotted). Datasets are randomly down sampled to 1,000,000 CpG positions covered by each (merged) dataset for density computation. **c** Histogram of deviations between matched CpG positions within the merged nanopore datasets and respective bisulphite orthologue for 5mC ($n = 15,468,504$ CpG sites) or 5hmC ($n = 11,283,580$). Deviations are expressed as the difference in the percentage of reads at any given base that are modified between techniques. Base-calls from matched CpG sites are merged across replicates. **d** Violin plots showing CpG modification rate Z-score across different genomic contexts. Each violin contains a boxplot representing the distribution of modification Z-scores for that feature; for each, black box shows the interquartile range, white centre line indicates the median, and black lines extending to the 1.5x interquartile range. Base-calls from matched CpG sites are merged across replicates. Y-axis is limited between $-3 \leq Z \leq 3$ for visualisation. Table S3 displays the summary statistics of Z-score and percentage modification for each feature.

## Table 2 | Intra-assay variation of each sequencing method

| | | Intra-assay RMSD at x depth (%) | | | Mean count of sites compared across replicates | | |
|---|---|---|---|---|---|---|---|
| | | 5x | 10x | 15x | 5x | 10x | 15x |
| Nanopore | 5mC | 17.8 | 16.8 | 15.3 | 24,840,932 | 19,873,534 | 9,765,218 |
| | 5hmC | 13.3 | 12.5 | 11.5 | 24,840,932 | 19,873,534 | 9,765,218 |
| oxBS-seq | | 20.0 | 14.1 | 8.4 | 1,838,247 | 22,292 | 3437 |
| TAB-seq | | 16.3 | 11.3 | 6.3 | 2,813,280 | 105,536 | 6126 |

Intra-assay variation shown for Nanopore sequencing ($N = 4$), oxBS-seq ($N = 2$), and TAB-seq ($N = 3$). Root Mean Square Deviation (RMSD) is calculated pairwise between all possible replicate pairs. The mean of these pairwise comparisons is shown here for different depths.

Intra-assay variation was calculated pairwise between replicates, using the root mean square of deviations (RMSD) between percentage modification values at matched CpG positions. The mean is then taken to summarise all pairwise comparisons. This value is expressed here as a percentage, reflecting deviations in the percentage of reads at any CpG sites that were modified. At 5x depth, we found mean intra-assay RMSDs of 17.8% for nanopore 5mC detection and 13.3% for 5hmC. At the same depth, intra-assay variation was slightly higher in oxBS-seq and TAB-seq, with mean RMSD values of 14.1% and 11.3% respectively. Using a sliding threshold on minimum coverage depth, we noted changes in RMSD as stricter thresholds were applied (Fig. S2a; Table 2). The oxBS-seq and TAB-seq data show lower rates of intra-assay variation at higher depth thresholds, appearing less variable; however, this likely reflects a dataset artefact. Few sites remain for comparison at these thresholds, leading to over-representation of a small subset of positions that have different rates of modification to the remainder of the sample (Fig. S2b).

Rates of modification detection at matched CpG sites are consistent between techniques (Fig. 2c). This inter-assay deviation is calculated using pairwise comparisons between replicates, calculating RMSD and Median Absolute Deviation (MAD). The mean of pairwise comparisons is reported here (Table 3). At 5x depth, we noted a MAD between 5mC sequencing

## Table 3 | Inter-assay variation between sequencing methods

| | Orthologous technique | MAD at x depth (%) | | | RMSD at x depth (%) | | |
|---|---|---|---|---|---|---|---|
| | | 5x | 10x | 15x | 5x | 10x | 15x |
| 5mC | oxBS-seq | 12.0 | 9.9 | 7.7 | 20.0 | 17.1 | 17.7 |
| 5hmC | TAB-seq | 7.6 | 7.0 | 4.7 | 15.8 | 12.5 | 9.9 |

Replicates of Nanopore sequencing ($N = 4$), oxBS-seq ($N = 2$), and TAB-seq ($N = 3$). Median Absolute Deviation (MAD) and Root Mean Square Deviation (RMSD) are calculated pairwise between all possible replicate pairs in both sequencing methods compared. The mean of these pairwise comparisons is shown here for different depths.

methods of 12.0% and RMSD of 20.0%. For 5hmC, compared to TAB-seq, MAD calculated to 7.6%, with an RMSD of 15.8%. As above, inter-assay deviation became lower with depth (Fig. S2c).

Rates of CpG methylation and hydroxymethylation are known to vary dependent on genomic context. To study this from these nanopore sequence data, CpG-context base-calls were aggregated according to overlapping genomic features, before a standard (Z)-score was calculated to indicate enrichment in a modification relative to the genomic mean. Within the nanopore data, patterns of 5mC and 5hmC enrichment across genomic features were typical of both modifications. Consistent with previous study

of these modifications, both 5mC and 5hmC are depleted in promoters (Fig. 1d; Table S3), 5' untranslated regions (5'UTRs), CpG islands (CGI), and "shore" regions in the 2 kb surrounding each CGI[44,45]. Unlike 5mC, which is enriched outside of genes, 5hmC is more abundant in genes, and is comparatively enriched in the "shelf" regions adjacent to CpG shores.

At genomic elements matched between the nanopore and TAB-seq datasets, we noticed that 5hmC Z-score correlates strongly between methods for each type of genomic feature (Fig. S3; Table S4), implying similar sensitivity to specific 5hmC-dense regions across the genome.

To compare larger scale modification trends, such as regional enrichment or depletion of 5hmC, we aggregated 5hmC base-calls within non-overlapping 500 bp tiles of the mouse genome. After calculating enrichment as before using Z-scores, we found a significant correlation in 5hmC Z-score between nanopore and TAB-seq (Spearman $\rho = 0.820$; $p < 0.0001$; $n = 881,755$). From this enrichment-focused approach we were interested to compare these tiles with public hMeDIP-seq data[46], a 5hmC-pulldown technique that detects 5hmC from "peaks" in coverage, representing high 5hmC signal. A large proportion of these peaks overlapped tiles enriched for 5hmC ($Z > 0$) in both the nanopore data (91.1%) (Fig. S4) and TAB-seq data (87.3% of peaks). Conversely, 57.3% of 5hmC-enriched peaks from the nanopore data (equivalent to almost 50 Mb) did not overlap any 5hmC peak regions. A weak monotonic relationship also exists between nanopore-detected 5hmC enrichment at these genomic windows and hMeDIP-seq peak fold enrichment (Spearman $\rho = 0.16$; $p < 0.0001$, $n = 274,286$), supporting the higher density of 5hmC detected by nanopore in these tiles.

### Direct sequencing of a 5hmC-enriched pulldown library
Given the apparent sensitivity of nanopore sequencing to 5hmC enrichment relative to both TAB-seq and conventional PCR-based hMeDIP-seq, as well as its reported efficacy even at ultra-low input concentration[47], we decided to assess the possibility of directly detecting base modifications from a 5hmC pulldown library. This would internally validate the enrichment of 5hmC by immunoprecipitation and test the sensitivity of the nanopore base-caller to altered modification levels.

We therefore sequenced the immunoprecipitation product of three hMeDIP reactions using an Oxford Nanopore Technologies MinION. Using MACS2, we identified a mean of 4654 peaks with an adjusted $p$-value per sample of $q < 0.05$ per run in an overall extremely shallow whole-genome dataset (0.005–0.01x depth) (Fig. 3a).

With the public hMeDIP-seq dataset previously used to compare enrichment over genomic windows as a reference, we found that, like conventional hMeDIP, our peaks were significantly more common within genic elements (binomial test; $p < 0.001$) and promoters relative to the background composition of the genome (Fig. 3b). Direct 5hmC base-calls within peak regions accounted for a significantly larger share (Welch's T-Test; $N = 3$; $p = 0.006$; Cohen's $d$: $d = 11.286164$) (26.4–31.4%) of CpG-context C base-calls than was detected by whole genome sequencing (9.6–10.3%) (Fig. 3c). 5mC was, by contrast, less abundant (41.4–47.6%) than in the whole genome (60.3–62.8%). 5hmC is densely distributed in most reads, with a mean of 0.29 5hmC base-calls per 100 bp.

5hmC was not detected in 34.6% of these sequence reads. This may represent nonspecific pulldown of DNA fragments low in 5hmC, namely at repetitive or high signal regions. Indeed, 5.5% of these 5hmC-negative reads overlapped regions of high signal listed in the ENCODE Blacklist[48], a further 12.7% intersect gamma satellite (GSAT) repeats, and 11.8% intersect simple repeat sequences identified by RepeatMasker[49]. As 5hmC is deposited in a strand asymmetrical manner[16,50], some could also represent the unmodified partners of hemi-hydroxymethylated CpG sites. Nevertheless, peak-calling using all reads supports the sensitivity of the nanopore base-caller to identify 5hmC-enrichment, with 93.0% of hMeDIP-seq peaks overlapping regions previously found to be enriched ($Z > 0$) for 5hmC in whole genome data (Fig. 3d). That a similarly high proportion of these peaks (92.1%) overlap with regions likewise enriched for 5hmC in TAB-seq suggests that these sequencing techniques are similarly sensitive to enriched hydroxymethylation.

Modified base detection in pulldown sequencing offers substantial utility in the ability to directly validate the presence or absence of base modifications, helping to inform against false positive enrichment peak detection. This also demonstrates the use of nanopore sequencing, paired with antibody enrichment, to directly detect base modifications in a pull-down library. This approach could be readily modified to target histone modifications and transcription factors, enabling direct identification of base modifications associated with these targets with ChIP-seq and CUT&RUN techniques in a benchtop context.

### Duplex sequencing detects CpG dyad asymmetry at molecular resolution
To this point, our data indicated that the nanopore base-caller is sensitive to 5hmC and functions consistently with other techniques. Using high duplex nanopore flow cells, we then aimed to investigate strand modification symmetry and asymmetry (Fig. 4a) with the ability to identify modification states present on both strands of a single DNA molecule, a level of detail not readily available with other techniques. This is done using primary or germline DMRs from a panel of imprinted genes, given the well-established stability of methylation at these loci[51].

Across the genome, duplex-paired reads accounted for a mean of 32% of sequence reads across eight sequencing replicates, including whole genome methylation controls (Fig. S5a). Of all CpG dyads, symmetrical modification (affecting both forward and reverse cytosine bases) states are the most abundant, with symmetrical 5mC accounting for 52.5% of all duplex base-calls and symmetrically unmodified C accounting for 23.5% (Fig. 4b; Fig. S5b, c). Hemi-modification, where the dyad includes a modified (5mC or 5hmC) base and an unmodified partner, accounts for a minority of duplex base-calls (C:5mC 6.3%; C:5hmC 2.0%). Here, "hetero-modified" refers to those CpG sites comprised of two different modified states, such as 5mC:5hmC pairs. Hetero-modified dyads are almost twice as prevalent as hemi-methylated dyads, accounting for 12.8% of all duplex modification patterns across all replicates. 5hmC is found predominantly in a hetero-modified state, being paired with 5mC in 72.3% of dyads containing 5hmC (Fig. S5d).

We isolated duplex reads from 15 previously defined DMRs (Table S5) and phased them into alleles according to modification state (Fig. 4c; Fig. S6)[52]. We found roughly equal ratios of methylated and unmethylated reads for most DMRs queried, as would be expected at imprinted loci (Fig. S7). Direct detection of 5hmC highlights distinct patterns of hydroxymethylation, and potentially demethylation, within individual alleles, such as patches of dense hydroxymethylation in the methylated alleles of *H19* and *Rasgrf1*, and groups of 5hmCpGs along the DMR boundaries of *Gnas1A* and *Igf2r* (Fig. S6).

Duplex patterns for symmetrical CpG dyads within these DMRs were significantly different from the genomic background (G-Test; $p < 0.001$) (Fig. 4d), consistent with DNA methylation being stably maintained at germline DMRs. Asymmetrical modifications (hemi-methylated and hetero-modified dyads), by contrast, are relatively rare, making up 7.7% of all duplex base-calls within these DMRs compared to the genome mean of 21.1% (Fig. S8). Comparing asymmetric CpG dyads in individual alleles, both the methylated allele and unmethylated alleles contained a smaller proportion of hemi-methylated (C:5mC) dyads compared to the genomic mean. Interestingly, hemi-hydroxymethylated (C:5hmC) dyads occurred with similar frequency on unmethylated alleles to the genomic mean.

### Symmetrically unmodified CpG dyads are enriched at CTCF binding sites
Having studied modification symmetry at imprinted DMRs, we selected CCCTC-binding factor (CTCF) as a test case for loci where symmetrical and asymmetrical strand modification are relevant for transcription factor binding[4]. Previously, CTCF was one of the first transcription factors shown to regulate genomic imprinting through its methylation sensitive binding to an imprinting control region and 3D organisation of DNA[53,54]. Recently, this has been shown to be sensitive to asymmetric strand modification, with different dyadic permutations found to have specific binding affinities for CTCF[4].

**Fig. 3 | Trial of direct nanopore sequencing of 5hmC-immunoprecipitated (IP) DNA.**
**a** Nanopore sequenced hMeDIP peaks overlapping the *Kcnj11* gene (chr7:45,746,000-45,751,000). For each repeat, tracks show the proportion of direct 5hmC calls at each CpG site (top) as well as sequencing coverage depth and the corresponding MACS2 narrow peak (bottom). Lowest tracks show WGS data from input. Produced using pyGenome-Tracks (v3.8)[93]. **b** Bar plot of primary genomic context overlapped by peak regions, comparing the direct nanopore hMeDIP ($N = 3$) with public hMeDIP-seq data ($N = 3$). Genomic background, based on mm39, provided as reference. Error bars indicate 1 standard deviation. **c** Bar plot showing modification states as a percentage of all CpG-context cytosine base-calls within the PromethION WGS data and hMeDIP peaks ($N = 3$). Error bars indicate 1 standard deviation. **d** Density plot underlay in shades of blue shows 5hmC Z-Score for matched 500 bp windows of WGS data from nanopore and TAB-seq. Z-scores are calculated using the arcsine transformed proportion of all CpG base-calls (merged across replicates) enclosed in a window detected as 5hmC. Peaks from all nanopore hMeDIP-seq replicates are overlaid onto the window they intersect as a scatterplot, with size and hue proportional to fold enrichment over input. Dotted line indicates sample mean 5hmC Z-Score for the (x) TAB-seq and (y) nanopore datasets. *$p < 0.05$; **$p < 0.01$; ***$p < 0.001$.

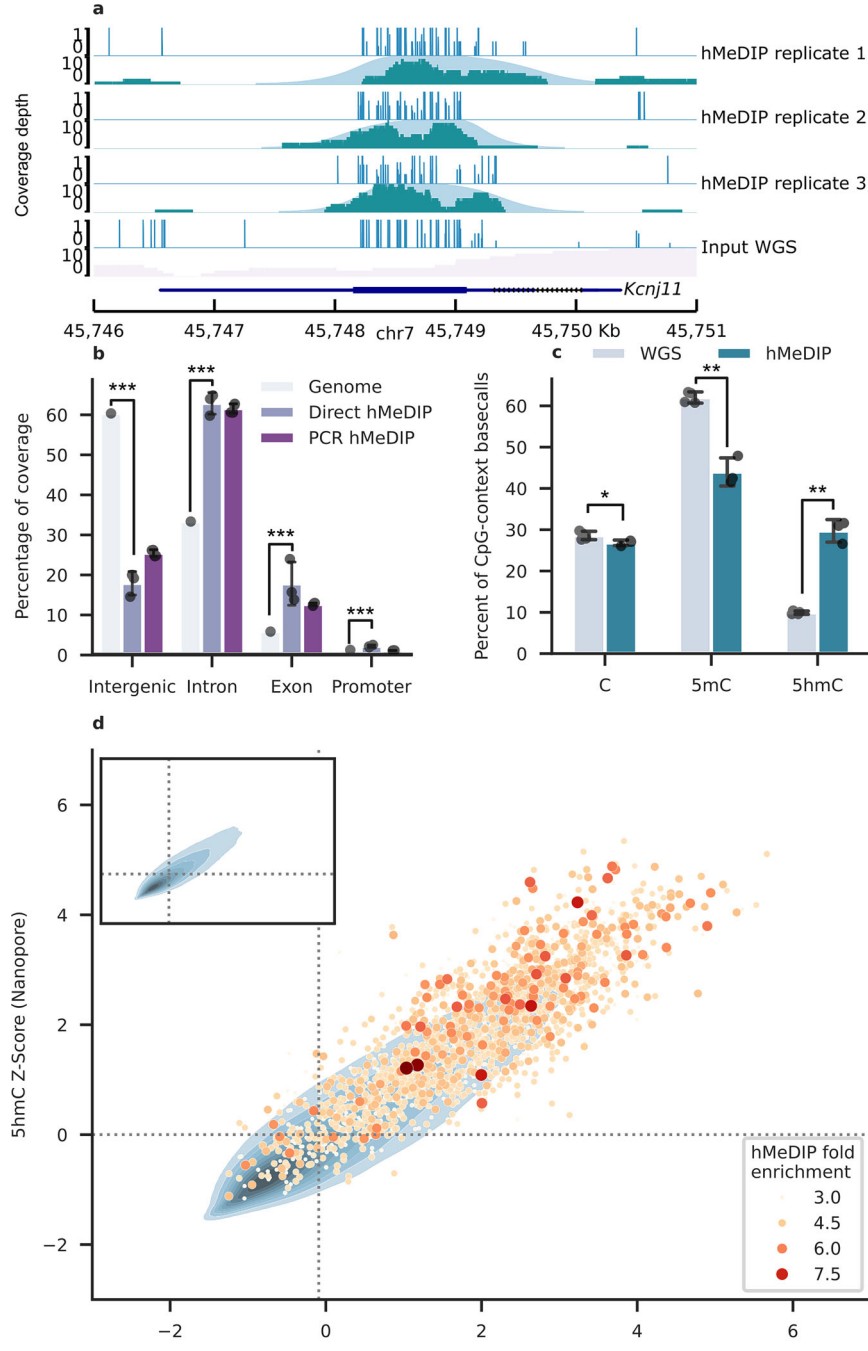

CTCF has specific affinity to a short and highly conserved sequence motif, a ~20-base sequence that often contains at least one CpG site. We located these motif sequences using the JASPAR MA0139.1 dataset[55], selecting motifs overlapping CpG sites. Using CTCF ChIP-seq data from ENCODE, we then found CpG sites present in motif sequences shown experimentally to bind CTCF in cerebellum. We then selected only those motif sequences overlapping ChIP-seq summits, representing the point of highest CTCF signal. Although duplex modification patterns for CpG sites present in CTCF sequence motifs were similar to the genomic background (Cramér's V: $\varphi_c = 0.11$), the modification patterns at CTCF-bound positions were very different ($\varphi_c = 0.65$). These bound motifs possess a significantly higher proportion of symmetrically unmethylated CpG base-calls (T-Test; $p<0.001$) (Fig. 5a), with corresponding depletion of all other dyad patterns.

Dyad modification states have a significant relationship with distance from CTCF ChIP-seq binding (Kruskal-Wallis H; $p<0.0001$), with a higher concentration of symmetrically unmodified C:C positions proximal to binding summits than all other dyad permutations (Fig. 5b). Only for this dyad type can a negative point-biserial correlation be found with absolute distance from binding summits ($r_{pb} = -0.32; p<0.0001$) (Table S6), implying closer proximity to CTCF-bound positions than other states in cerebellum. Dyads with symmetrically modified 5mC:5mC ($r_{pb} = 0.25; p<0.0001$) and asymmetrical hetero-modified 5mC:5hmC ($r_{pb} = 0.11; p<0.0001$) are positively correlated with distance from binding summits, concurring with the known methylation sensitivity of CTCF. Both hemi-modified C:5mC and C:5hmC, as well as symmetrically hydroxymethylated 5hmC:5hmC positions, have a very weak correlation with distance to ChIP-seq summits.

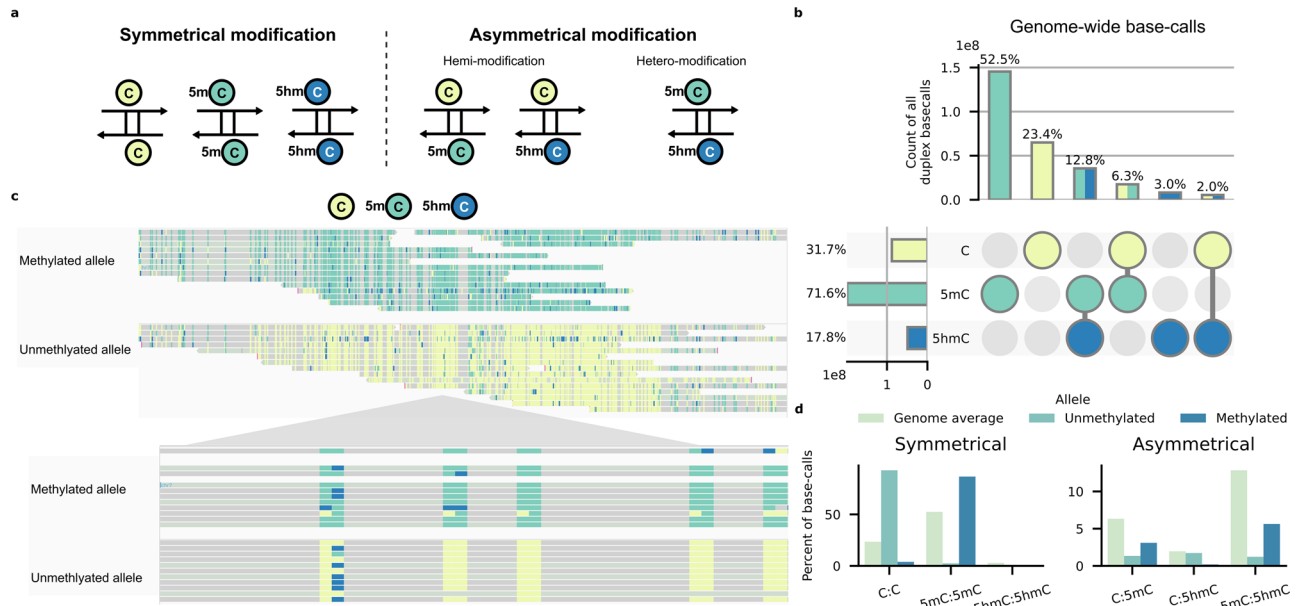

**Fig. 4 | Duplex detection of modified bases using High Duplex PromethION flow cells.** Reads from all replicates are concatenated here for visualisation. **a** Schematic summary of currently detectable modification states from duplex reads. **b** UpSet plot of CpG-context cytosine base-calls from all duplex reads. Produced using UpSetPlot (0.9.0)[94]. Top: percentage of duplex reads described by the intersection matrix below. Below: percentage of duplex reads containing one or more of a given cytosine state. **c** IGV view of the Nespas-Gnasxl imprinted locus from nanopore duplex sequencing. Reads are separated by allele and have been down sampled for visualisation. Above: overview of the whole imprinted locus (chr2:174,134,000-174,145,000). Below: close-up IGV view of duplex modified base-calls over a 50 bp segment. Duplex modified bases appear as blocks of two consecutive modification calls, with the forward (top) modification indicated by the colour on the left and the reverse (bottom) modification indicated by the colour on the right. **d** Comparison of the mean percentage of all duplex modified base-calls in the whole genome, as in (**b**), to modified base-calls within the methylated and unmethylated alleles of 15 imprinted loci. Duplex base-calls across replicates are merged.

Asymmetrical methylation, with 5mC opposite the motif strand, was previously found to be more favourable to CTCF binding than symmetrically unmethylated CpG sites[4]. In our data, asymmetrically modified CpG sites comprise only a small percentage (4.6%) of dyad base-calls at bound CTCF positions, with unmodified cytosine present on the motif strand in 56.2% of those cases.

The orientation of CTCF binding sites has been shown to be important for loop formation, with loops favouring convergently oriented CTCF motifs[56]. In mouse, 9373 CTCF motifs are palindromic, potentially enabling bidirectional CTCF binding from either or both strands. We found 1365 (14.6%) palindromic CTCF motifs that directly intersect CTCF ChIP-seq peaks (Fig. 5c). Asymmetrical methylation at these motifs is uncommon in our duplex data however, occurring at least once in only 188 (13.8%) bound motifs. The 102 genes associated with these motifs are most enriched for genes involved in nervous system (Benjamini-Hochberg $p = 0.0039$), whole body (Benjamini-Hochberg $p = 0.0039$), and brain (Benjamini-Hochberg $p = 0.0097$)[57,58].

We hypothesise that strand asymmetry at palindromic CTCF motif sequences could influence the orientation of CTCF binding (Fig. 5d). In such a model, 5mC present on the forward strand at these palindromic positions would favour loop formation in a reverse direction, whereas 5mC on the reverse strand would favour loops in the opposite direction. Symmetrically unmodified CTCF motifs enable bidirectional binding of CTCF and enable looping in both or either direction. Future research, pairing nanopore duplex sequencing of a CTCF ChIP library and HiC data, would test this model and any effects this may have on gene expression.

## Discussion

Cytosine methylation has important biological and evolutionary roles; however, the roles of higher oxidation states such as 5hmC, formed via the action of TET enzymes, are less well understood. Here we demonstrate that the detection of 5mC and 5hmC by nanopore sequencing is consistent with widespread techniques and opens new modes of studying these modifications with known involvement in disease, ageing, and development.

The DMRs of imprinted genes represent loci with known stable methylation across most somatic tissues. We have demonstrated the use of direct modification detection from nanopore sequence data to study strand modification symmetry and allele-specific methylation at DMRs in a panel of imprinted genes. This confirmed previous findings suggesting high methylation fidelity at germline DMRs[59,60], observing depletion of asymmetrical strand modification states on both the methylated and unmethylated allele compared to the genomic background. 5hmC is present on both alleles in all DMRs, though is depleted relative to the genomic background. Our study did not examine secondary DMRs, which may be less stable and thus have more variable methylation states in mouse brain. Secondary DMRs have previously been shown to have higher amounts of asymmetric modifications[59]. Future studies examining the extent of hetero-modifications, potentially distinguishing transient and stable states of asymmetric modifications and 5hmC, could determine whether these states play a role in maintaining normal methylation patterns at imprinted as well as other loci.

Long read sequencing data have previously been used to identify and phase patterns of allele-specific methylation[61–63], with longer reads serving to bridge regions of low SNP density and offering greater utility to haplotyping tools. In the inbred mouse strain used in this study, where no genomic variation could be anticipated between parental alleles, we capitalised on the allele-specificity of methylation at imprinted genes to separate alleles. The quantitative ability of nanopore sequencing is also highlighted at imprinted loci in the equal ratios of methylated to unmethylated reads at most DMRs. For the discovery of new imprinted loci or types of allele-specific modifications, where haplotype construction is required, long reads with direct modification readouts will greatly simplify haplotype reconstruction, where haplotype information can become masked by bisulphite conversion itself.

Although 5hmC, and later oxidation derivatives of 5mC, have been shown to be deposited in a largely strand asymmetrical manner[16,50,64], it is interesting to find such a high degree of asymmetrically hydroxymethylated positions in cerebellum, a terminally differentiated tissue. Unlike the canonical cytosine base and its methylation derivative 5mC, symmetrically

**Fig. 5 | Duplex base-calling at CTCF-binding sites.**
**a** CpG dyad duplex modification pattern across the whole genome, CTCF binding motifs, and ChIP-seq peak summits (split into two plot areas for visibility of dyad states with a small proportion of base-calls). Error bars indicate 1 s.d. Replicates are shown as separate dots ($N = 4$). **b** Violin plot representation of duplex modification pattern distance to CTCF ChIP-seq summit sites. CpG dyads more than 500 bp from a summit are not represented. Duplex base-calls from all replicates are merged. **c** IGV view of *F830045P16Rik* locus (chr2: 129,338,223-129,385,892) highlighting two CTCF ChIP-seq. peaks overlapping palindromic CTCF motif positions. **d** Schematic representation of methylation and strand symmetry dependent CTCF loop formation. 1: A forward strand CTCF motif (left) overlaps a symmetrically unmodified CpG site and faces an asymmetrically methylated palindromic CTCF motif (right). Methylation on the forward strand of the palindromic motif sequence favours CTCF binding in a reverse orientation, convergent with the previous unmodified motif and enabling loop formation. 2: Symmetrical modification of a CpG site (left) prevents CTCF binding and forbids loop formation. 3: Inversion of strand modification asymmetry at the palindromic motif (right) favours CTCF binding in a forward orientation. **\*\****p* < 0.01; **\*\*\****p* < 0.001; **\*\*\*\****p* < 0.0001.

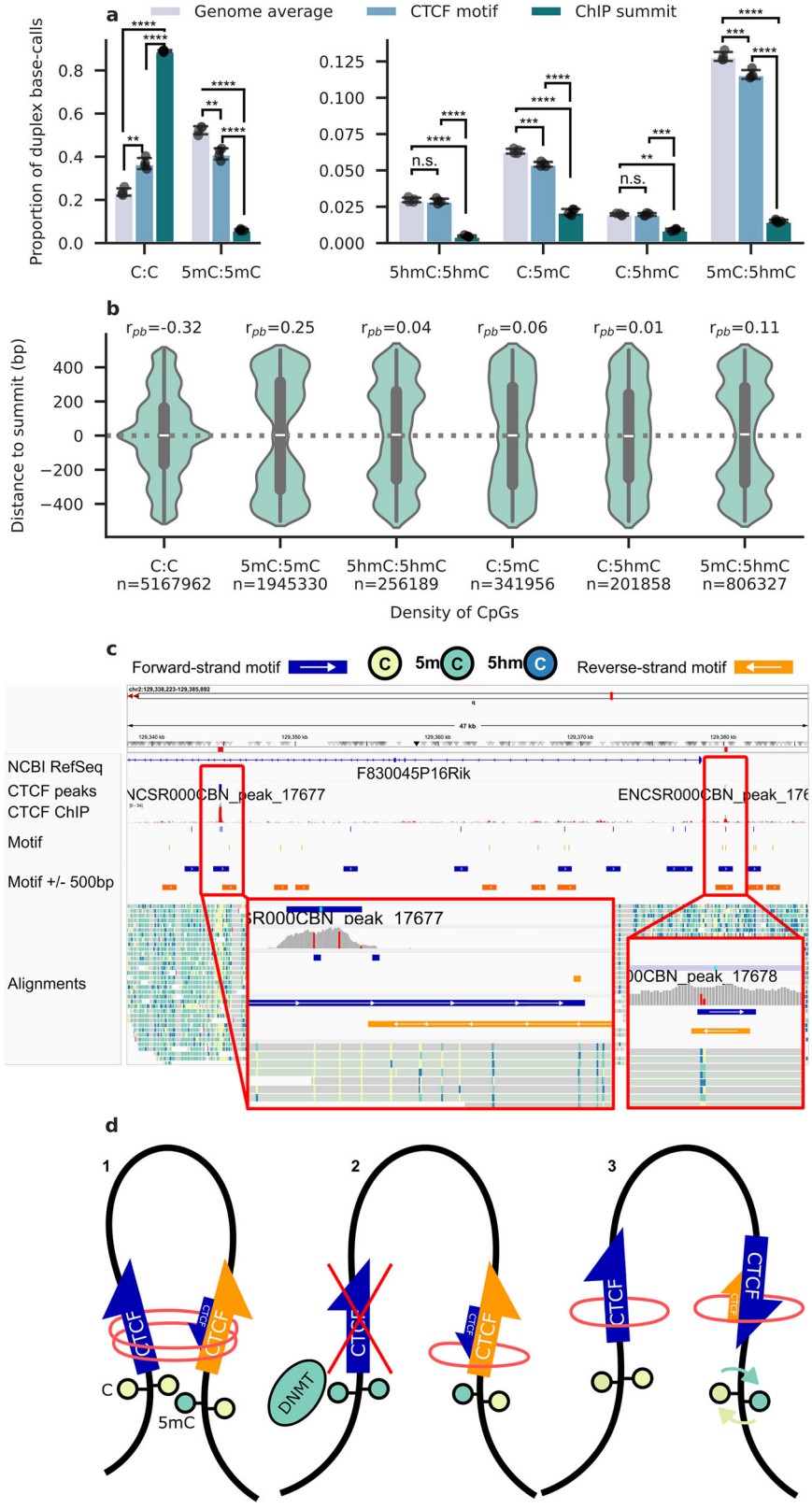

modified 5hmC:5hmC positions were rare, confirming previous findings that the TET enzymes function in a primarily strand specific manner[50]. Given that 5mC is largely symmetrically deposited between both strands at the CpG site, the presence of asymmetrical hemi-hydroxymethylated C:5hmC pairs is especially surprising, which could be evidence of incomplete demethylation.

To test the use of duplex modified base detection in a biologically meaningful context, we compared our duplex sequence data with CTCF ChIP-seq data from a similar biological sample, finding patterns of strand modification symmetry and asymmetry unlike the genomic background. These results confirm the preference for unmodified cytosine for CTCF binding. In future, direct nanopore sequencing of a CTCF-ChIP pulldown

library, using duplex reads, would add further sensitivity to the detection of asymmetric modifications and CTCF binding.

There are some technical considerations for experimental design using duplex sequencing for sequencing asymmetrical CpG modification, as it takes two simplex reads to get a single duplex image of a CpG dyad. Duplex capture is still only moderately effective in High Duplex flow cells, with a mean of 32% of reads found in duplex pairs. If duplex capture is the objective, aiming for total coverage at least 6–7-times higher than target duplex depth is advisable. As an example, median genomic coverage in our murine samples was 29-33x, whereas mean dyad depth in duplex was 2–2.5x.

It remains to be seen whether future developments will include increased rates of duplex read capture. This, along with improved base calling algorithms to enable detection of later TET-mediated oxidation products of 5mC, and other base modifications including 6mA, would provide a great advantage to epigenetic research, culminating in the detection of a complete vocabulary of DNA modifications, detectable on both strands of the same molecule, as part of standard sequencing practice.

In summary, nanopore sequencing enables robust direct detection of three epigenetic states of cytosine in DNA: C, 5mC, and 5hmC. This method provides several advantages compared to other commonly used techniques, allowing high throughput direct and accurate readout of modification states on both strands of a single DNA molecule as part of standard sequencing practice.

## Methods

### Publicly available data benchmark data
Data for 5mC and 5hmC were procured from the oxBS-seq and TAB-seq experiments, respectively, performed by Ma et al. on cerebella from three 8-week-old female C57BL/6 mice using Illumina HiSeq X Ten (data available on the Genome Sequence Archive of the Beijing Institute of Genomics under experiment identifiers CRX008031 and CRX008032)[42,65,66]. These were downloaded in fastq format and aligned to the mm39 (GRCm39) mouse reference genome using Bowtie2 and Bismark[67,68].

hMeDIP-seq data from homogenous mouse tissues produced by Song et al.[46], stored under Gene Expression Omnibus (GEO) accession GSE25398, were downloaded in fastq format. Quality control was performed using TrimGalore! (0.6.10)[69], before alignment to mm39 reference using Bowtie2[68]. Duplicate sequence reads were marked using Picard MarkDuplicates[70]. Regions intersecting the ENCODE blacklist file for mm39[48] were removed using bedtools (v2.31.1) intersect[71], before peak calling was performed using MACS2[24] with options: "--gsize 'mm' --call-summits".

An identical pipeline[72] was used for CTCF ChIP-seq and input sequence data produced by the Bing Ren Lab of UCSD, stored in the ENCODE Portal under accession codes ENCSR000CBN and ENCSR000CAT, respectively[73]. These were produced from cerebellum tissues derived from two 8-week-old male C57BL/6NCrl mice.

### Animal samples and DNA extraction
No statistical calculation was used to predetermine sample size, which was instead designed to optimise sequencing depth. Brain tissues were obtained from two 8-week-old female C57BL/6NCrl (Charles River Laboratories) mice procured from the University of Bath animal facility. The project was given a favourable opinion by the University of Bath's Animal Welfare and Ethical Review Body (AWERB; Review Reference 3436-4022), a Committee defined in law under the UK Animals (Scientific Procedures) Act, and approved by the University's Academic Ethics and Integrity Committee (AEIC). We have complied with all relevant ethical regulations for animal use.

High molecular weight genomic DNA was extracted from these cerebella using the QIAGEN MagAttract HMW DNA Kit (QIAGEN, 67563) using a modified protocol available from the Oxford Nanopore Technologies Community website[74]. Quality control was performed on a DeNovix DS-11 FX+ Spectrofluorometer.

### Whole genome sequencing of mouse cerebellar gDNA
Cerebellar gDNA was sheared using a Covaris g-TUBE (Covaris, 520079) to a target length of 8 kb. Whole genome sequencing (WGS) libraries were prepared in duplicate for each biological replicate, using the Ligation Sequencing Kit V14 standard protocol (Oxford Nanopore Technologies, SQK-LSK114) with reagents from the NEBNext® Companion Module for Oxford Nanopore Technologies® Ligation Sequencing (New England Biolabs, E7180S). Libraries were quantified using a Qubit 4 fluorometer (ThermoFisher Scientific, Q33238) with Qubit dsDNA HS (ThermoFisher Scientific, Q33230). 150 ng of each sample was loaded onto PromethION (Oxford Nanopore Technologies, PRO-SEQ048) flow cells (Oxford Nanopore Technologies, FLO-PRO114HD). All sequencing runs were manually stopped after 72 h. Each run generated 65–80 Gb over 9–15 million sequence reads.

### Whole genome sequencing of human methylated/unmethylated controls
Two libraries were prepared as positive and negative methylation controls using the Human Methylated & Non-Methylated (WGA) DNA Set (Zymo Research, D5013). Libraries were quantified using a Qubit 4 fluorometer (ThermoFisher Scientific, Q33238) with Qubit dsDNA HS (ThermoFisher Scientific, Q33230). Sample DNA was sheared to 8 kb, and library preparation was performed in duplicate using the Ligation Sequencing Kit V14 standard protocol (Oxford Nanopore Technologies, SQK-LSK114). Libraries were sequenced by loading 150 ng onto PromethION (Oxford Nanopore Technologies, PRO-SEQ048) flow cells (Oxford Nanopore Technologies, FLO-PRO114HD), with runs lasting 72 h. Runs produced 25–52 Gb over 5–9 million reads.

### Nanopore hMeDIP-seq
DNA from mouse cerebellum was sheared to a target of 6 kb using a Covaris g-TUBE (Covaris, 520079). An aliquot was sequenced as a control on an R10.4.1 MinION flow cell (Oxford Nanopore Technologies, FLO-MIN114) using Ligation Sequencing (SQK-LSK114) Kit 14 following the associated protocol. Sheared DNA was sonicated in triplicate using a Bioruptor® Pico sonication device (Diagenode, B0106001) in a 1.5 mL Bioruptor® Pico Microtube with cap (Diagenode, C30010016) to target fragment lengths <500 bp. A 2200 TapeStation system (Agilent, G2991A) with D1000 ScreenTapes (Agilent, 5067-5582) and associated reagents (Agilent, 5067-5583) was used to assess the fragment size distribution of each replicate (mean 362 bp).

5hmC immunoprecipitation was performed based on Nestor and Meehan[23]. 1 μg sonicated dsDNA was incubated overnight at 4 °C rotating in IP buffer (10 mM Na-phosphate pH 7.0, 140 mM NaCl, 0.05% Triton X-100) with either 1 μL (1:500) of whole serum α-5hmC antibody (ActiveMotif, 39769; lot: 23720003) or 2 μL (1:250) IgG-purified α-5hmC antibody (ActiveMotif, 39791; lot: 25419010) per reaction. Samples were then incubated with 40 μL equilibrated Dynabeads Protein G (Invitrogen, 10003D) for 1 h while rotating at 4 °C and washed three times in ice-cold IP buffer for 5 min at room temperature. Bound DNA was eluted in digestion buffer (50 mM Tris-HCl pH 8.0, 10 mM EDTA, 0.5% SDS) and 200 ng proteinase K (ThermoFisher Scientific, EO0491) for 3 h at 50 °C on a shaker at 800 rpm. Immunoprecipitated DNA was purified using the QIAQuick PCR Purification kit (QIAGEN, 28104) following manufacturer instructions and eluted into 50 μL elution buffer before sequencing.

The eluate was prepared for sequencing following the MinION protocol SQK-LSK114. Libraries were loaded onto three R10.4.1 MinION flow cells and sequenced, generating 154-382 Mb of sequence data per flow cell.

### Base-calling and data exclusion
Base-calling was performed using the Oxford Nanopore Technologies open-source Dorado base-caller (v0.5.1). For the PromethION data, the super-accurate (sup@v4.3.0) base-calling model was used with 5mC and 5hmC base-calling (5mCG_5hmCG@v1). The hMeDIP-seq data were produced earlier, using base-calling sup@v4.2.0. Minimap2 (v2.24-r1122) was used to produce an index file of the mm39 (GRCm39) mouse reference

genome[75]. Aligned sequence data was then sorted and indexed using the samtools sequence alignment tool suite (samtools 1.19)[76]. Modified bases were extracted from these reads using 'modkit pileup' for simplex and 'modkit pileup-hemi' for duplex basecalls (0.2.8)[77]. The options '--only-tabs --cpg' and '--mask' were used against the mm39 mouse reference genome fasta file from UCSC. The '--mask' argument excludes regions that intersect those flagged by the RepeatMasker software to reduce the influence of highly repetitive elements[49]. Reads are extracted using 'modkit extract' with the '--read-calls-path <PATH > --cpg --mask' arguments.

CpG sites were filtered to a minimum sequencing depth of 5x. Data points below this depth are excluded.

At each CpG site, the ratio of sequence reads in which the C base is modified to all C calls from that site (i.e., excluding C-to-N SNPs) is computed to summarise its level of modification. Using 5hmC as an example:

$$\%_{5hmC} = \frac{5hmC}{5hmC + 5mC + C} \times 100$$

## GC content definition

In both the human (hg38/GRCh38) and mouse reference genomes (mm39/GRCm39), mean GC content is calculated using 'bedtools nucBed'[71], which summarises the total number of G and C bases in all chromosomes in the reference genome. Mean genomic GC is then calculated as the sum of G and C bases divided by the total length of all non-N bases.

To determine local GC content, bigwig files of GC content over 5 bp windows was downloaded for the respective genome from the UCSC Genome Browser[78]. 100 bp bed intervals were defined using the bedtools command 'makewindows'. This bed file was used together with the UCSC tool bigWigAverageOverBed (v2) with the '-bedOut' option to calculate the mean GC content over those 100 bp intervals. Data from CpG sites from the nanopore data were overlapped to these 100 bp intervals. Finally, all CpG sites were grouped into 5-percentile bins of GC content ranging from 0 to 100%, producing ($n = 20$) bins.

## Gene and CpG island definition

Gene bodies are defined using the GENCODE Consortium Basic annotation (VM32)[79]. Promoter regions are defined as regions starting 1 kb upstream of a gene. CpG islands (CGI) were downloaded as a track from UCSC Table Browser for mm39[78]. Shores are defined as 2 kb regions immediately up- and downstream of CGI, and shelves are defined as regions 2–4 kb up- and downstream of CGI following Rechache et al.[80]

## Nanopore hMeDIP-seq peak calling and comparison to WGS

Whole genomic DNA from the same sample, without treatment, was sequenced on a MinION flow cell for use as an input sample. The Model-Based Analysis of ChIP-Seq (MACS2) software (v.2.2.6) was used to call narrow peaks relative to this input with the '--gsize mm' option[24]. From the resulting narrow peaks, those overlapping regions on the ENCODE blacklist were removed[48]. By default, 'MACS2 callpeaks', peaks are filtered out if they have an adjusted p-value (Benjamini-Hochberg corrected) greater than 0.05.

For comparison with WGS, Nanopore and TAB-seq derived WGS data were tiled into non-overlapping 500 bp windows. Windows containing fewer than 10 CpG sites were discarded. The overlap between narrow peaks and these genomic tiles was then found using PyRanges[81]. Where peaks overlapped with multiple windows, the window with the highest degree of overlap was selected. Only windows containing at least half of an overlapping peak region were analysed.

## DMR methylation clustering in duplex reads

Germline DMR coordinates were obtained from Tomizawa et al.[52] These were lifted over to mm39 using the UCSC liftOver tool[82]. After visualisation, some coordinates have been manually shifted to apparent boundaries of the differentially modified region. After defining these as regions of interest, duplex reads overlapping these regions were extracted from the alignment files using 'samtools view' with '--tag dx:1', from which read-level information was extracted for CpG sites using 'modkit extract' with the '--cpg' flag and the '--reference' and '--include-bed' arguments.

Allelic phasing was performed in Python: CpG sites on each read were nominally encoded by modification state. Sites absent from a read are also encoded. Reads containing too high a proportion of absent positions are excluded. Each read is therefore represented by a string of CpG site modification states, which were then used to produce a Hamming distance matrix. The mean distance between reads, and then successive clusters of reads, produced a hierarchical clustering pattern, which was flattened to the two highest cluster levels. Read IDs extracted from these clusters were then used for 'samtools view' and the '--qname-file' argument to extract reads into new bam files.

## Statistics and reproducibility

The number of replicates (N) and observations (n) used in tests are provided in text. Test statistics, p-values, and effect sizes, are provided rounded to two significant figures. Additional details, including exact $p$ values and unrounded test statistics, are provided in full in Supplementary Data 1.

Mean rates of CpG modification are compared across four replicates of nanopore data and three replicates of either oxBS-seq or TAB-seq data, for 5mC and 5hmC respectively, using a two-tailed Welch's T or unequal variances test, given differences in both sample size and sequencing depth. Root mean square deviation (RMSD) is calculated to compare percentage modification rates at matched CpG sites between datasets of the same sequencing method, as a metric for intra-assay variation, and as an inter-assay metric for CpG level deviation differences between the nanopore, oxBS-seq, and TAB-seq datasets. This is calculated for all replicate pair permutations, from which the mean is taken as a summary statistic.

To compare modification detection at intervals spanning multiple CpG sites, such as genomic features, genes, or genomic windows scale; the count of modified base detections is taken across the interval, along with the total sum of all CpG-context base-calls. The ratio between the two is taken to find the proportion of all CpG base-calls within the interval with a given modification. Arcsine transformation of these proportions is used to approximately normalise the distribution of proportion values and a standard score (Z-score) is calculated from transformed values.

Comparing the proportion of coverage made up of given features in direct nanopore hMeDIP-seq with the genomic background is performed using one-tailed binomial tests with $n = 17,368$ total observations from all replicates. The genome was divided into non-overlapping 500 bp windows, with the proportion of these primarily overlapping a given type of feature used to inform $p$ for each binomial test. The count of peaks that primarily overlap a given feature informs $k$.

Comparing the proportion of CpG-context base-calls comprised of a given state is performed using Welch's T, with three replicates of nanopore hMeDIP-seq ($N = 3$) compared to four replicates of nanopore WGS ($N = 4$).

Comparison of duplex modification states between DMRs and the genomic mean is performed using a G-test, equivalent to a chi-square test performed using a log-likelihood ratio to account for very large count values. The expected frequency of duplex patterns used for this test is derived from proportion of duplex modification states in the given state from over WGS replicates.

The level of association between CpG dyad modification patterns between the whole genome, CTCF binding motif sequences, and CTCF motif sequences overlapping ChIP-seq summits, is expressed using Cramér's V or $\varphi_c$, where high values of $\varphi_c$ are taken to suggest greater dependence between context (i.e., CTCF motif, or motif at summit) and dyad modification pattern. Paired two-tailed T-tests with $N = 3$ are used to compare individual differences in the proportions of duplex CpG base-calls made up of each dyad state between the whole genome, CTCF motifs, and motifs at binding summits. These are paired on account the fact that CpG sites at bound summits are a subset of those at all CTCF motifs, with both being a subset of the CpG sites included in the whole genome.

The Kruskal–Wallis test is used to determine whether there is a statistically significant difference between different CpG dyad modification states and absolute distance to CTCF ChIP-seq summits. This result is followed by post hoc analysis using Dunn's test with a Holm-Bonferroni

correction on calculated $p$ values. To further assess the relationship between dyad modification state and absolute distance to ChIP summit, point biserial correlations are calculated for each state.

Tissue enrichment for asymmetrically methylated CTCF binding sites is calculated using the STRING database[58], using a Benjamini-Hochberg corrected $p$ value threshold of 0.05.

Statistical analyses were largely performed using Python. Annotations are given within scripts where necessary to improve reproducibility or indicate where any non-Python code is used.

## Reporting summary
Further information on research design is available in the Nature Portfolio Reporting Summary linked to this article.

## Code availability
Data analysis was performed in Python and Jupyter Notebooks using additional open-source software packages, including pandas (2.0.1)[83], numpy (1.24.3)[84], scipy (1.13.1)[85], scikit-learn (1.5.2)[86], PyRanges (0.0.120)[81], and Pingouin (0.5.4)[87]. Visualisations were produced using matplotlib (3.7.1)[88], seaborn (0.13.2)[89], and the Integrative Genomics Viewer (IGV) (2.19.1)[90]. Other graphing packages are mentioned in text where used. Scripts and analysis code are publicly available via a project GitHub repository (v1.0.0, DOI: 10.5281/zenodo.14753748)[91]. An executable pipeline, ChIP2MACS2 (v1.0.0, https://doi.org/10.5281/zenodo.14535833), which includes Bowtie2, TrimGalore!, and MACS2 and related tools, was also produced to handle hMeDIP-seq/ChIP-seq alignment and peak calling pipeline used in these analyses[72].

## Data availability
Raw nanopore machine data in *fast5* format has been made available for all murine whole genome sequence experiments on the Sequence Read Archive (SRA) as BioProject PRJNA1144670. Aligned sequence data in BAM file format is also available. Additionally, CpG context modified base detections, as produced by 'modkit pileup' are available on the NCBI GEO archive with the accession: GSE279860. These data are limited to CpG positions relative to the mm39 reference genome and are soft-masked. Machine data is not available for the Zymo DNA Methylation Standards; however, these are available as BAM format files under the same BioProject. For the nanopore hMeDIP-seq experiments, data is available in both BAM format under the previously mentioned SRA BioProject, as well as in *pod5* format on Zenodo, with record https://doi.org/10.5281/zenodo.14514705. BAM format sequence data used as an input is available as SRR30150148 on the SRA. Narrow peak data, along with direct modified base detections from those peaks, is downloadable under the GEO Series GSE288331. Source data for figures can be downloaded from Figshare (https://doi.org/10.6084/m9.figshare.28287962.v2)[92].

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

## Acknowledgements

This work was supported by funding from the Engineering and Physical Sciences Research Council (EPSRC) [grant number 2598658]. Oxford Nanopore Technologies provided financial support for nanopore sequencing. In addition, the authors would like to thank Sarah Corsi, Nathan Bagby, Marcus Stoiber, Mark Bruce, and Adrien Leger from Oxford Nanopore Technologies for their invaluable support and recommendations. The authors would like to thank Dr. Kim Moorwood of the University of Bath for her support in procuring biological samples for this research. F.H. is supported by the Biotechnology and Biological Sciences Research Council-funded South West Biosciences Doctoral Training Partnership (DTP3: BB/T008741/1) in partnership with CASE partner bit.bio.

## Author contributions

D.O.H. performed nanopore sequencing experiments. D.O.H. wrote scripts to analyse data and produce figure panels. F.H. designed and performed immunoprecipitation experiments. D.O.H., S.B., and A.M. wrote the report. S.R. reviewed analyses and provided guidance on statistics. A.M conceived and supervised the project. All authors have reviewed results and commented on the manuscript.

## Competing interests

The authors declare the following competing interests: D.O.H. receives support from Oxford Nanopore Technologies as the CASE Studentship partner for his doctoral degree. This support has included financial support for Oxford Nanopore Technologies consumables, use of facilities, and sponsored access to training.
