## [Transparent Peer Review file · Communications Biology]

Double and single stranded detection of 5-methylcytosine and 5-hydroxymethylcytosine with nanopore sequencing

Corresponding Author: Mr Dominic Halliwell

Version 0:

Reviewer comments:

Reviewer #1

(Remarks to the Author)

The authors have provided a comprehensive study of the application of ONT Nanopore sequencing for investigating 5mC and 5hmC DNA modifications, contrasted against existing short read bisulfite sequencing. They also assessed the use of duplex reads to investigate strand asymmetric modifications, using CTCF binding sites in mouse cerebellum as a test case. The manuscript represents an important step in the understanding of ONT Nanopore capabilities and will be particularly useful to anyone wanting to investigate the role of strand asymmetric DNA modifications. The depth of genomic coverage that is demonstrated in this study, and the detailed description of preparation methods enables the authors to improve the level of reporting across loci without "breaking the bank", which should encourage further researcher interest.

The statistical tests applied here appear sensible and well chosen, and the level of documentation of study methods is excellent.

The authors need to make style adjustments to match the journal standards - references, figure legends, text colour and fonts all need adjustment.

Reviewer #2

(Remarks to the Author)

This manuscript presents a potentially significant contribution to field by validating the use of the use of nanopore sequencing to profile DNA methylation and DNA hydroxymethylation, generating novel data from mouse cerebellum samples. Currently, the field lacks a comprehensive assessment of the ability to accurately call 5hmC and there is a lot of data here to address that gap. However, for this study to fully benefit the scientific community, a more in-depth analysis and critical discussion are necessary.

While the findings are promising, the organisation of the results could improved to provide a stronger narrative. The frequent changes between different sample types, techniques and analyses make it challenging to follow and ultimately synthesise the results. More attention should be given to outline the purpose of each analysis before the result is presented and link the analyses together into a cohesive presentation.

The specific points I outline below primarily concern the statistical analyses and their reporting. I have provided suggestions aimed at improving how the manuscript is presented facilitating more informative synthesis of the findings.

Comments

1. I would provide the summary of the fully methylated and unmethylated control samples, as a distinct section first rather than intersect it with the cerebellar samples. Especially as they are from a different species. I would be interested in the 5hmC levels for these samples. Where it gets it wrong does it typically predict the methylated control as C or 5hmC or for unmethylated control what is the breakdown of 5mC vs 5hmC calls? Have these analyses been performed before, how do your false positive and false negative rates compare to those reported previously? Finally, are these perfect controls? Should they have all cytosines methylated or unmethylated such that we can interpret the error rates as absolute values? Where there were errors, where they random or can we learn anything about where these errors typically occur? i.e. certain sequence contexts? In the main text you refer to replicates but from the results it is unclear how these were used as just a single statistic is reported.
2. Reference is made to replicates, but there no description at the beginning of the results as to how many samples were processed and whether the data presented represents one sample or multiple samples collapsed together. It is unclear if, and which point, replicates and even samples were combined. Is it not always clear what the comparisons are. Where you

are comparing replicates within Nanopore data, and where you are comparing across technologies. In the latter situation, what did you do with the replicates? This can be improved by providing more narrative to the objective of each analysis before describing the results with links made to draw the results into a coherent message.

3. In the results, please include the summary statistics of the mean read depth that substantiate the comparative statement, rather than just reference to a supplementary figure.

4. Average is a vague term, it can be the mean or median etc. Please reconsider if a more accurate term can be used.

5. Can you hypothesise why the drop in coverage around CpG islands/promoters is smaller for nanopore sequencing? Has this been documented before? Strikes me as strong positive for nanopore sequencing.

6. Figure 1a and 1b are not the typical way to present the distribution of 5mC or 5hmC. Can you include a standard distribution of the levels of 5mC and 5hmC across the genome? I would like to see the bimodal distribution of 5mC.

7. This statement didn't make sense to me:

"62.7% of CpG site detections consisting of 5mC on average, across replicates of the nanopore sequence data"

Do you mean that 62.7% of CpG sites had at least 1 read of 5mC called?

8. I am surprised that the difference between technologies (62.7% vs 71.2%) was not highly significant given the number of CpG sites you have data for. Especially given the results for 5hmC, with a much smaller difference, is significant? I assume you did it across replicates/samples, hence you had a small n? You also don't refer this test result in the text, what is the interpretation/value of this result in the context of this study?

9. You report the RMSD as a percentage, is this a true %, or a consequence of the fact that you have quantified modifications as a %? i.e. you reported statistic that the value of 16.79% for nanopore 5mC detection means that is it within 16.79% of the actual value (i.e. if the true value was 16.79% the nanopore value is typically between 14.4-19.6%) or there is difference of 16.79% for nanopore 5mC detection (i.e. if the true value was 16.79% the nanopore value is typically between 0-33.6%).

10. The finding that 5hmC is found predominantly in a hetero-modified state, this is consistent with 5hmC being either a false negative for 5mC (or vice versa) and 5hmC being an intermediate state following demethylation (i.e. not stable).

11. Are results driven by higher coverage in nanopore data. For example, for the RMSD calculations across replicates for the different technologies, where nanopore is superior does that result hold if the coverage was more equal? What if you limit comparisons to sites where the oxBS or TAB-seq data had more reads? Or downsample nanopore data?

12. Figs S1a-c, are these across all replicates? Given the comparisons between replicates for each technology, important to understand how coverage varies across replicates?

13. Care is needed when interpreting the difference in correlation statistics between 5mC and 5hmC across assays. Due to the large range of values for 5mC, the correlation is likely to be higher driven by the bigger variation. Also in this section, for 5mC you report the "median absolute difference" and for 5hmC you report the "median deviation" between technologies, why the inconsistency or is this a typo?

14. Figures 1d-f, would be helpful to have some sense of the distribution or variance rather than just the means. Could these be violin plots/boxplots? I initially struggled to synthesise Figures 1d and 1e, but I think that is because the scales on the y axis are dramatically different. The difference is much smaller than the difference between genomic features shown in 1d. I think the addition of the variance or some more formal statistics that account for this will help contextualise the meaning of this difference.

15. What was the purpose of the tiling approach? What does this add to the comparison at a CpG site level? I observe a much higher correlation at this level between technologies at the CpG site? What can you learn from this? How did you define a 5hmC window as enriched? Is this the binomial test that is described in the methods? If so, what p-value threshold was used. Why did you not consider flipping the comparison around and quantifying what hMeDIP misses? i.e. how many enriched windows do not overlap with peaks?

16. In this sentence the results of the test (p-value, effect size) are missing: "Direct 5hmC base-calls within peak regions accounted for a significantly larger share (Welch's T-Test; N=3) (26.5-31.4%) of CpG-context C base-calls than was detected in WGS (9.5-10.4%) (Fig. 2c). 5mC was, by contrast, less abundant (41.3-47.6%) than in WGS (60.3-63.3%)."

17. Are the rates of asymmetrical modifications, consistent with the error rates you find? In other words as the 5mC/5hmC capture is not 100%, you would expect some true symmetrical modifications to be misread as asymmetrical proportion to the false negative rate. Do you find statistical evidence of increased rates of these relative to what you would expect by chance?

Reviewer #3

(Remarks to the Author)

This article performs benchmarking of ONT nanopore sequencing in detecting 5mC/5hmC modifications, further investigating the unique ability of nanopore sequencing to detect cytosine modifications in higher resolution. Especially, the authors explore how duplex sequencing can help analyzing symmetry and asymmetry modifications, by investigating strand-specific modifications in DMRs and CTCF sites. To my knowledge, this is the first study that performs systematic benchmarking and analyses of nanopore sequencing for detecting 5hmC and hemi-methylation. However, I still have some concerns.

This study uses a lot of technical replicates for each sequencing technique. However, I cannot find any basic statistics of all used sequencing data (e.g., mean/median genome coverage, mean/median read length) in either the manuscript or the supplementary, which I think is important to show.

- Following this concern, in Supplementary Figure 1a-c, does the histogram show information of all replicates or only one of them?

- Figure 1a-b shows that nanopore profiles much more cytosines than oxBS and TAB. Is this mainly because the genome coverage of the oxBS/TAB data are not enough?

- Does the authors have any suggestions on how many coverage of nanopore reads is enough for performing strand-specific

modification analyses?

It seems that the sample used for generating oxBS/TAB data is not exact the same as the sample used for nanopore sequencing. It would be better if the samples are the same for generating nanopore and oxBS/TAB reads, for benchmarking.

Regarding data availability, it would be nice that the authors make their POD5 data publicly available.

Page 11 Line 337, is "(Table S2)" should be "(Table S3)"?

Version 1:

Reviewer comments:

Reviewer #1

(Remarks to the Author)

The additions and amendments made by the authors have resulted in a more informative and easier reading manuscript. The clarification of reported statistics and data sources is much appreciated. In particular the new figures seem very clear and comprehensive. Thank you for your hard work with the revisions.

A suggested modification for clarity:

Line 102: METEORE 102

was later developed as a consensus approach combining predictions from nanopore, DeepSignal 103 (Ni et al., 2019), DeepMod (Liu, Q. et al., 2019), Oxford Nanopore Technologies' Guppy, Megalodon, 104 and Tombo (Yuen et al., 2021)

It looks like Tombo is attributed to Yuen et al, rather than METEORE. It might be best to show attribution of Guppy, Megalodon and Tombo to ONT, and put the Yuen citation immediately after METEORE.

Reviewer #2

(Remarks to the Author)

I am happy with the responses.

Reviewer #3

(Remarks to the Author)

I thank the authors for their hard work. All of my previous concerns have been addressed. I now only have one minor concern about the data availability. According to the authors' response, I am not sure if they uploaded all the raw data to NCBI, as I just saw four tar.gz files, but there are more than 10 nanopore sequencing runs in the Bioproject. I suggest the authors describe more clearly about the raw reads they have uploaded, if they could. Overall, I recommend this manuscript to be accepted by Communications Biology.

#	Reviewer Comment	Response
1	The authors have provided a comprehensive study of the application of ONT Nanopore sequencing for investigating 5mC and 5hmC DNA modifications, contrasted against existing short read bisulfite sequencing. They also assessed the use of duplex reads to investigate strand asymmetric modifications, using CTCF binding sites in mouse cerebellum as a test case. The manuscript represents an important step in the understanding of ONT Nanopore capabilities and will be particularly useful to anyone wanting to investigate the role of strand asymmetric DNA modifications. The depth of genomic coverage that is demonstrated in this study, and the detailed description of preparation methods enables the authors to improve the level of reporting across loci without "breaking the bank", which should encourage further researcher interest. The statistical tests applied here appear sensible and well chosen, and the level of documentation of study methods is excellent. The authors need to make style adjustments to match the journal standards - references, figure legends, text colour and fonts all need adjustment.	Reply: We thank this reviewer for their time and feedback. Style adjustments have been made.
2	This manuscript presents a potentially significant contribution to field by validating the use of the use of nanopore sequencing to profile DNA methylation and DNA hydroxymethylation, generating novel data from mouse cerebellum samples. Currently, the field lacks a comprehensive assessment of the ability to accurately call 5hmC and there is a lot of data here to address that gap. However, for this study to fully benefit the scientific community, a	Reply: We thank the reviewer for their thoughtful comments and feedback.

	more in-depth analysis and critical discussion are necessary. While the findings are promising, the organisation of the results could improved to provide a stronger narrative. The frequent changes between different sample types, techniques and analyses make it challenging to follow and ultimately synthesise the results. More attention should be given to outline the purpose of each analysis before the result is presented and link the analyses together into a cohesive presentation. The specific points I outline below primarily concern the statistical analyses and their reporting. I have provided suggestions aimed at improving how the manuscript is presented facilitating more informative synthesis of the findings.	
3	I would provide the summary of the fully methylated and unmethylated control samples, as a distinct section first rather than intersect it with the cerebellar samples. Especially as they are from a different species. I would be interested in the 5hmC levels for these samples. Where it gets it wrong does it typically predict the methylated control as C or 5hmC or for unmethylated control what is the breakdown of 5mC vs 5hmC calls? Have these analyses been performed before, how do your false positive and false negative rates compare to those reported previously? Finally, are these perfect controls? Should they have all cytosines methylated or unmethylated such that we can interpret the error rates as absolute values? Where there were errors, where they random or can we learn anything about where these errors typically occur? i.e. certain sequence contexts? In the main text you refer to replicates but from the results it is unclear how these were used as just a single statistic is reported.	Reply: We apologise for this oversight and have now included a section describing the control samples and rates of error. Actions:  • We have added a new first section (lines 126-174) (copied below table) with a new figure (Fig. 1). • Figure number has been adjusted accordingly. • Earlier interpretation of error rates using these controls has been replaced: (Lines: 215-218) “False negative 5mC detection is unlikely to fully account for this, with the difference between means (6.0%) larger than that expected from the false negative error (3.0%) noted above, potentially indicating a bias favouring the detection of methylated bases in oxBS-seq.” (Lines: 220-224) “In the absence of a ground-truth 5hmC control, this difference (difference between means: 1.6%) may approximate the rate of false negative error for 5hmC detection by nanopore sequencing. 5hmC was significantly more likely to be detected at any given CpG position at least once, likely due to the

		difference in depth (Welch's T-Test; $p=0.01$; Cohen's $d: d=3.21$)."
4	Reference is made to replicates, but there no description at the beginning of the results as to how many samples were processed and whether the data presented represents one sample or multiple samples collapsed together. It is unclear if, and which point, replicates and even samples were combined. Is it not always clear what the comparisons are. Where you are comparing replicates within Nanopore data, and where you are comparing across technologies. In the latter situation, what did you do with the replicates? This can be improve by providing more narrative to the objective of each analysis before describing the results with links made to draw the results into a coherent message.	Reply: We apologise for this oversight and have improved the narrative around replicates in text. Overall, two duplicates were used of both the modification negative DNA standard and modification positive standard. Two biological replicates were sequenced in duplicate for whole mouse genome Nanopore sequencing. Where replicates are compared, as in the case of RMSD calculations within (intra-assay) and across technologies (inter-assay), this is performed by taking all possible pairs of replicates in a given comparison (e.g., from the Nanopore data and TAB data). These statistics are then summarised using a mean. Actions:  • We have changed the beginning of that section in the results (lines: 191-200) with the following text to clarify what replicates were made and why they were used. "To assess modified base detection from ex vivo tissues and compare rates of base modification with orthologous techniques, cerebellar tissues from two 8-week-old female mice were sequenced in duplicate, producing four whole genome datasets (median genomic depth 29-32x) (Table S1). CpG modification calls were extracted from this, with median CpG depths of 15-17x (stranded) (Fig. S1a). Samples were selected to match a publicly available archive of sequence data, containing whole genome oxidative bisulphite sequencing (oxBS-seq) for two biological replicates and TET-assisted bisulphite sequencing (TAB-seq) for three biological replicates (Ma et al., 2017). The datasets produced using these techniques were lower coverage depth, with a median CpG depth of 3x (stranded) for TAB-seq (Fig. S1b), and 2-6x for oxBS-seq (Fig. S1c). To account for higher rates of FPR found in some repeat contexts, repeats were soft-masked."  • Edited figure/table legends clarify where replicates are separate, and how merging was done if not. Figure 2: Line 179.

		Figure 3: Lines 295, 298, 300-301. Figure 4: 340-341, 348, 350. Figure 5: 388, 390. Fig. S1: 3-4. Fig. S3: 23 Fig. S4: 28-29. Fig. S5: 37-40. Fig. S6: 43. Fig. S7: 46-47. Fig. S8: 50-51.  • We have clarified in text where and how replicates are compared in pairwise comparisons: (Lines: 227-229) “Intra-assay variation was calculated pairwise between replicates, using the root mean square of deviations (RMSD) between percentage modification values at matched CpG positions. The mean is then taken to summarise all pairwise comparisons.” (Lines 244-246): “This inter-assay deviation is calculated using pairwise comparisons between replicates, calculating RMSD in the same manner as above, and Median Absolute Deviation (MAD). The mean of pairwise comparisons is reported here (Table 3).”  • A description is added in Statistics and Reproducibility on how these pairwise RMSD calculations are produced (lines 640-647).
5	In the results, please include the summary statistics of the mean read depth that substantiate the comparative statement, rather than just reference to a supplementary figure.	Reply: We agree with the reviewer that these summary statistics are important and have summarised these statistics in text as requested. In addition, we have provided full median depth statistics in a new supplementary table. (Table S1). Actions:  • Summary statistics are added to the text: (Lines 193-200): “CpG modification calls were extracted from this, with median CpG depths of 15-17x (stranded) (Fig. S1a). Samples were selected to match a publicly available archive of sequence data, containing whole genome oxidative bisulphite sequencing (oxBS-seq) for two biological replicates and TET-

		assisted bisulphite sequencing (TAB-seq) for three biological replicates (Ma et al., 2017). The datasets produced using these techniques were lower coverage depth, with a median CpG depth of 3x (stranded) for TAB-seq (Fig. S1b), and 2-6x for oxBS-seq (Fig. S1c). To account for higher rates of FPR found in some repeat contexts, repeats were soft-masked.” (Lines 201-203): “Higher genomic depth in the nanopore sequence data (Fig. S1d) provides a larger sample of sequence reads per cytosine (Fig. S1e).”  • A new table has been added (Table S1, below) with summary statistics on sample depth.
6	Average is a vague term, it can be the mean or median etc. Please reconsider if a more accurate term can be used.	We apologise for any lack of clarity and have amended this term accordingly. Actions:  • We have removed or replaced all instances of “average” in text and figures.
7	Can you hypothesise why the drop in coverage around CpG islands/promoters is smaller for nanopore sequencing? Has this been documented before? Strikes me as strong positive for nanopore sequencing.	Reply: Thank you for pointing this out. Previous publications have identified depth of coverage biases in bisulphite techniques related in large part to bisulphite conversion, selective fragment degradation, as well as PCR amplification (Olova et al. 2018). This has the effect of over-representation in regions of high methylation, and under-representation at regions such as promoters and CGI. As GC richness is found to be a primary driver of this effect, since writing, we have analysed the effect of GC on coverage depth in each dataset to better address this comment. Nanopore sequencing is less vulnerable to GC% biases, with a drop in coverage depth only in contexts that have substantially higher GC% than the mean (41.7%), whereas oxBS-seq and TAB-seq drop in coverage even in regions below genomic mean GC%. Actions:  • We have added the following to line 202-203: “Previous studies in bisulphite based libraries have highlighted a negative correlation between GC content and sequencing depth (Olova et al., 2018).”

		Full reference: Olova, N., Krueger, F., Andrews, S., Oxley, D., Berrens, R.V., Branco, M.R. and Reik, W., 2018. Comparison of whole-genome bisulfite sequencing library preparation strategies identifies sources of biases affecting DNA methylation data. Genome Biology, 19(1), p. 33.  • We have added plots to Figure S1 demonstrating coverage depth as a function of GC context. • We have interpreted these findings as follows: (Lines 203-209): “In our study we noted significant negative correlation with depth (Fig. S1f) and local GC content for all methods (Nanopore: $\rho=-0.53, p\leq 0.001$; oxBS-seq: $\rho=-0.71, p\leq 0.001$; TAB-seq: $\rho=-0.79, p\leq 0.001$), with oxBS-seq and TAB-seq showing the most prominent loss of coverage even below the genomic mean GC content (41.7%) (Nanopore: $\rho=-0.54, p\leq 0.001$; oxBS-seq: $\rho=-0.71, p\leq 0.001$; TAB-seq: $\rho=-0.59, p\leq 0.001$). Thus GC-related effects may explain the lower depth of coverage found in promoters and CpG islands (Fig. S1h), as well as general loss of coverage (Fig. S1i).”
8	Figure 1a and 1b are not the typical way to present the distribution of 5mC or 5hmC. Can you include a standard distribution of the levels of 5mC and 5hmC across the genome? I would like to see the bimodal distribution of 5mC.	Reply: Thank you for this feedback. Actions:  • We have replaced the cumulative distribution plots used previously with two kernel density estimate plots accordingly (see Figure 2, below table). • The following in-text description has also been added: (Lines: 210-212): “Modified base detection is similar in the nanopore sequencing and bisulphite-based data. CpG methylation in either method follows a bimodal distribution (Fig. 2a), with two maxima of CpG positions that are either completely methylated or completely unmethylated.” (Lines: 219-220): “5hmC is unimodally distributed around 0% in either method, with a maximum reflecting a high density of CpG sites in which 5hmC is not found (Fig. 2b).”

9	This statement didn't make sense to me: "62.7% of CpG site detections consisting of 5mC on average, across replicates of the nanopore sequence data" Do you mean that 62.7% of CpG sites had at least 1 read of 5mC called?	Reply: We apologise for this confusion. This has been amended. Actions:  This has been corrected and amended (see response to Comment 10)
10	I am surprised that the difference between technologies (62.7% vs 71.2%) was not highly significant given the number of CpG sites you have data for. Especially given the results for 5hmc, with a much smaller difference, is significant? I assume you did it across replicates/samples, hence you had a small n? You also don't refer this test result in the text, what is the interpretation/value of this result in the context of this study?	Reply: Thank you for highlighting this. Originally, this calculation was performed using data previously filtered on depth. On reflection, we have since recalculated the proportion of all bases detected as modified using the raw modified base count instead, giving a more complete representation of modified base detection by each method. For nanopore, 60.3-62.8% of all CpG-context base-calls are methylated, and 66.3-68.8% of base-calls in oxBS-seq. With these data we found the same test outcomes, with a non-significant difference between the total proportion of base modifications detected by Nanopore and oxBS-seq: TtestResult (statistic=-4.121977129536045, pvalue=0.08021051597269349, df=1.5707965724244235). The reviewer is likely correct that it is the small sample size (Nanopore (N=4) and oxBS-seq (N=2)) responsible for the non-significant result. There remains a significant difference compared with TAB-seq despite the small numerical differences: TtestResult(statistic=-5.3837633443237625, pvalue=0.004111732265761028, df=4.4914773631538125). This could likewise be a possible effect of higher sample size, with N=3 for the TAB-seq data. Actions:  We have provided revised statistics in table form (Table S2, below) and have updated the T-test statistics in text (lines: 214, 222). Updated test statistics in Supplementary Statistics. This has been updated in text to expand on this (lines: 212-215): "Fewer 5mC detections are made by the nanopore base-caller (60.3-62.8%) than by oxBS-seq (66.3-

		68.8%) (Table S2); however, the difference is non-significant (Welch's T-Test: $p=0.08$; Cohen's $d: d=4.06$), potentially on account of sample size differences."
11	You report the RMSD as a percentage, is this a true %, or a consequence of the fact that you have quantified modifications as a %? i.e you reported statistic that the value of 16.79% for nanopore 5mC detection means that is it within 16.79% of the actual value (i.e. if the true value was 16.79% the nanopore value is typically between 14.4-19.6%) or there is difference of 16.79% for nanopore 5mC detection (i.e. if the true value was 16.79% the nanopore value is typically between 0-33.6%).	Reply: We apologise for the confusion here. The reviewer is right, this is not a true percentage and is indeed a consequence of the fact that modifications are quantified as percentages. Actions:  This was added to improve clarity (lines 229-230): "This value is expressed here as a percentage, reflecting deviations in the percentage of reads at any CpG sites that were modified."
12	The finding that 5hmC is found predominantly in a hetero-modified state, this is consistent with 5hmC being either a false negative for 5mC (or vice versa) and 5hmC being an intermediate state following demethylation (i.e. not stable).	Reply: Interesting point. It is not known whether the predominance of hetero-modified states containing 5hmC is indicative of instability and have therefore refrained from speculating on this. However, we can address the possibility of false detection of 5hmC. We have produced new statistics for the error rates of 5mC and 5hmC detection in response to Comment 3. We have found that false positive 5hmC detection occurs at a rate of 1.34% (1.1% + 0.24% in ground-truth methylated and ground-truth unmethylated positions respectively) of all CpG base-calls. Given that a CpG dyad is comprised of two cytosine base-calls, up to 2.68% (1.34% x 2) of all dyads may contain at least one of these false positive 5hmC base-calls. The share of all duplex base-calls comprised of hetero-modified positions (12.8%) is almost 5-times larger, therefore it is unlikely that false positive error accounts for many of these observed asymmetrical sites.
13	Are results driven by higher coverage in nanopore data. For example, for the RMSD calculations across replicates for the different technologies, where nanopore is superior does that result hold if the coverage was more equal? What if you limit comparisons to sites where the oxBS or TAB-seq data had more reads? Or downsample nanopore data?	Reply: Thank you for these suggestions. It is not feasible to down-sample our nanopore data due to the tabulated format of the nanopore modified base output. If we were to do this, it would introduce random variance that we would struggle to distinguish from genuine differences in modified base detection.

Therefore, we have incorporated this reviewer's other suggestion to sample sites where oxBS-seq and TAB-seq had more reads. We used a sliding minimum depth threshold – selecting sites with at least 5x depth per replicate, 10x, and 15x, and so on, calculating RMSD at each level of depth. As expected, we found that higher depths corresponded with lower rates of inter- and intra-assay deviation; however, this may be an artefact of the unequal coverage within the bisulphite sequencing datasets. In this case, thresholding led to the sampling of a decreasingly small subset of over-represented positions – seemingly with different rates of modification to the mean.

Actions:

- We have made coverage depth thresholds equal and changed the corresponding line in Materials and Methods:

(line 581):

“CpG sites were filtered to a minimum sequencing depth of 5x.”

- We have produced a new supplementary figure (Fig. S2, below) comparing sequencing depth and inter and intra-assay RMSD.
- This is interpreted in text as follows:

(Lines 230-238):

“At 5x depth, we found mean intra-assay RMSDs of 17.8 % for nanopore 5mC detection and 13.3% for 5hmC. At the same depth, intra-assay variation was slightly higher in oxBS-seq and TAB-seq, with mean RMSD values of 14.1 % and 11.3% respectively. Using a sliding threshold on minimum coverage depth, we noted changes in RMSD as stricter thresholds were applied (Fig. S2a; Table 2). The oxBS-seq and TAB-seq data show lower rates of intra-assay variation at higher depth thresholds, appearing less variable; however, this likely reflects a dataset artefact. Few sites remain for comparison at these thresholds, leading to overrepresentation of a small subset of positions that have different rates of modification to the remainder of the sample (Fig. S2b).”

(Lines 246-248):

“At 5x depth, we noted a MAD between 5mC sequencing methods of 12.0% and RMSD of 20.0%.”

		For 5hmC, compared to TAB-seq, MAD calculated to 7.6%, with an RMSD of 15.8%. As above, inter-assay deviation became lower with depth (Fig. S2c).”  Statistics have been added in table form as Table 2 and Table 3.
14	Figs S1a-c, are these across all replicates? Given the comparisons between replicates for each technology, important to understand how coverage varies across replicates?	Reply: We apologise for the lack of clarity here. These plots only presented the merged depth of each dataset. This figure has now been amended to communicate coverage across replicates in response to this comment. Actions:  This supplementary figure, Fig. S1, has been changed to clarify the distribution of depth and coverage across sequencing replicates of each technique (Fig. S1; below). Text for lines 191-200 was changed in response to Comment 4
15	Care is needed when interpreting the difference in correlation statistics between 5mC and 5hmC across assays. Due to the large range of values for 5mC, the correlation is likely to be higher driven by the bigger variation. Also in this section, for 5mC you report the “median absolute difference” and for 5hmC you report the “median deviation” between technologies, why the inconsistency or is this a typo?	Reply: Thank you for this note. We agree with this. There are indeed several limitations in using correlation statistics in this context. As the reviewer noted, there are differences in variance between rates of CpG methylation and hydroxymethylation, and in addition there are a high proportion of tied ranks (repeated observations of 0% 5hmC, 100% 5mC) in either dataset, making the correlation statistics less robust. As these results could be misleading, we have removed both from the Results. Regarding the inconsistency with “median deviation”; this was a typo and should be “median absolute deviation”. Actions:  We have removed the correlation statistics and interpretation. Typo has been corrected (line 247).
16	Figures 1d-f, would be helpful to have some sense of the distribution or variance rather than just the means. Could these be violin plots/boxplots? I initially struggled to synthesise Figures 1d and 1e, but I think that is because the scales on the y axis are dramatically different. The difference is much smaller than the difference between	Reply: Thank you for this feedback; we have made the changes suggested. Actions:  Figure has been amended based on this feedback (see Figure 2, below table). The adjoining text (lines 254-263) has been replaced to better interpret the changed figure.

	genomic features shown in 1d. I think the addition of the variance or some more formal statistics that account for this will help contextualise the meaning of this difference.	“Rates of CpG methylation and hydroxymethylation are known to vary dependent on genomic context. To study this from these nanopore sequence data, CpG-context base-calls were aggregated according to overlapping genomic features, before a standard (Z)-score was calculated to indicate enrichment in a modification relative to the genomic mean. Within the nanopore data, patterns of 5mC and 5hmC enrichment across genomic features were typical of both modifications. Consistent with previous study of these modifications, both 5mC and 5hmC are depleted in promoters (Fig. 1d; Table S3), 5’ untranslated regions (5’UTRs), CpG islands (CGI), and “shore” regions in the 2kb surrounding each CGI (Madrid, Chopra and Alisch, 2018; Wilkins et al., 2020). Unlike 5mC, which is enriched outside of genes, 5hmC is more abundant in genes, and is comparatively enriched in the “shelf” regions adjacent to CpG shores.”  • A new supplementary table (S3, copied below table) has been added to provide summary statistics for each context type.
17	What was the purpose of the tiling approach? What does this add to the comparison at a CpG site level? I observe a much higher correlation at this level between technologies at the CpG site? What can you learn from this? How did you define a 5hmC window as enriched? Is this the binomial test that is described in the methods? If so, what p-value threshold was used. Why did you not consider flipping the comparison around and quantifying what hMeDIP misses? I.e. how many enriched windows do not overlap with peaks?	Reply: The purpose of the tiling approach was to investigate whether larger scale modification trends, such as regional enrichment or depletion in 5hmC, would be consistent across sequencing techniques. We were motivated by the stochastic model of DNA methylation proposed by Jeltsch and Jurkowska (2014), which discuss regional and cellular factors (e.g., local enzyme activity, duplication rate) as determinative of modification probability. Within regions, the higher combined depth provides a better predictor of modification probability than at the level of a single CpG site, particularly in the shallower depth TAB-seq data, where only a small random sample of reads were captured. We interpreted the higher correlation we found as evidence that the modification probabilities we detected are broadly similar to those in oxBS-seq and TAB-seq. Enrichment was defined relative to the genomic mean (represented by $Z = 0$). Sites with $Z \geq 0$ were considered to be enriched and sites with $Z \leq 0$ as depleted. This was not calculated using a binomial, and was simply based on this threshold.

		Thank you for the suggestion regarding the hMeDIP-seq comparison. Although 91.0% of peaks were in windows found to be enriched by WGS, the proportion of enriched windows that overlapped a peak was much lower, at 42.7%; thus 57.3% of enriched windows do not overlap a hMeDIP-seq peak. Our intention in the former comparison was to note the proportion of regions we found to be enriched by WGS that likewise show up in hMeDIP-seq. We did not think it fair to compare the opposite however. hMeDIP-seq is not suited as a whole genome sequence method and pulls down only 5hmC-enriched fragments, but we're unaware of how enriched these fragments need to be. Given the threshold for enrichment of $Z > 0$, this could lead to misleading comparison where the precision of this threshold and the hMeDIP-seq are not the same. Actions:  • Tile enrichment is clarified: Line 273-274: "A large proportion of these peaks overlapped tiles enriched for 5hmC ($Z > 0$) in both the nanopore data (91.1%) (Fig. S4) and TAB-seq data (87.3% of peaks)." Line 321: "overlapping regions previously found to be enriched ($Z > 0$) for 5hmC in WGS data (Fig. 3d)."
18	In this sentence the results of the test (p-value, effect size) are missing: "Direct 5hmC base-calls within peak regions accounted for a significantly larger share (Welch's T-Test; N=3) (26.5-31.4%) of CpG-context C base-calls than was detected in WGS (9.5-10.4%) (Fig. 2c). 5mC was, by contrast, less abundant (41.3-47.6%) than in WGS (60.3-63.3%)."	Reply: Thank you for noting this error. We have corrected this in text. Actions:  • Added statistics to line 308: "(Welch's T-Test; N=3; $p=0.006$; Cohen's $d=11.286164$) (26.4-31.4%)"
19	Are the rates of asymmetrical modifications, consistent with the error rates you find? In other words as the 5mC/5hmC capture is not 100%, you would expect some true symmetrical modifications to be misread as asymmetrical proportion to the false negative rate. Do you find statistical evidence of increased rates of these	Reply: The rates of asymmetrical modification detected are consistent with the rates of false positive/negative error. We include a calculation below to illustrate this point. Example: The false positive rate for 5hmC is 0.24% for a single ground truth C base. In a symmetrical dyad comprised of two ground truth C bases, the

	relative to what you would expect by chance?	probability of at least one false 5hmC detection is therefore 0.48%. 23.4% of all dyads were detected in a symmetrical C state; therefore, we would expect at least one false positive 5hmC detections in at least 0.11% all CpG dyads (0.0048×0.234), resulting in an apparent C:5hmC pair. By contrast, this state was detected in 2% of all dyads; therefore, the difference (1.89%) provides statistical evidence of these sites being detected above that expected by chance error alone. This is not including compounded error rates from errors in both partners, the chance of which is negligibly small.
20	This article performs benchmarking of ONT nanopore sequencing in detecting 5mC/5hmC modifications, further investigating the unique ability of nanopore sequencing to detect cytosine modifications in higher resolution. Especially, the authors explore how duplex sequencing can help analyzing symmetry and asymmetry modifications, by investigating strand-specific modifications in DMRs and CTCF sites. To my knowledge, this is the first study that performs systematic benchmarking and analyses of nanopore sequencing for detecting 5hmC and hemi-methylation. However, I still have some concerns.	Reply: We thank the reviewer for taking the time to leave this feedback.
21	This study uses a lot of technical replicates for each sequencing technique. However, I cannot find any basic statistics of all used sequencing data (e.g., mean/median genome coverage, mean/median read length) in either the manuscript or the supplementary, which I think is important to show. Following this concern, in Supplementary Figure 1a-c, does the histogram show information of all replicates or only one of them?	Reply: We apologise for the lack of clarity and have made changes to include this information in response to other comments (Comments 4, 5). Actions:  • We have produced a new table (Table S1) and Supplementary Figure (Fig. S1). • We have amended this supplementary figure to show coverage depth and breadth within each replicate (Fig. S1a-c).
22	Figure 1a-b shows that nanopore profiles much more cytosines than oxBS and TAB. Is this mainly because the genome coverage of the oxBS/TAB data are not enough?	Reply: Yes, this is correct; more cytosine bases were covered in the Nanopore datasets. This is in large part due to coverage thresholding, where a large number of positions in the oxBS/TAB data were excluded for being too shallow (depth < 5x).

23	Does the authors have any suggestions on how many coverage of nanopore reads is enough for performing strand-specific modification analyses?	Reply: We have assumed this comment refers to strand analysis using duplex sequencing. Firstly, each duplex “read” is comprised of two simplex reads paired together informatically. As a result, it takes two simplex reads to get a single duplex image of a CpG dyad. Duplex capture is still only moderately effective in these “High Duplex” flow cells we used. Across all sequencing runs, a mean of 32% of our reads were in duplex pairs, and the remainder were simplex reads where the other simplex partner was not captured. Standard commercially available flow cells have a lower capture rate (in our experience, around 10%). We would thus recommend, if duplex capture is the objective, aiming for total coverage at least 6-7-times higher than target duplex depth. As an example, median genomic coverage in our murine samples was 29-33x, whereas mean dyad depth in duplex was 2-2.5x. Action:  We have added a comment to this effect in the discussion (lines 484-490): “There are some technical considerations for experimental design using duplex sequencing for sequencing asymmetrical CpG modification, it takes two simplex reads to get a single duplex image of a CpG dyad. Duplex capture is still only moderately effective in High Duplex flow cells, with a mean of 32% of reads found in duplex pairs. If duplex capture is the objective, aiming for total coverage at least 6-7-times higher than target duplex depth is advisable. As an example, median genomic coverage in our murine samples was 29-33x, whereas mean dyad depth in duplex was 2-2.5x.”
24	It seems that the sample used for generating oxBS/TAB data is not exact the same as the sample used for nanopore sequencing. It would be better if the samples are the same for generating nanopore and oxBS/TAB reads, for benchmarking.	Reply: This is correct; the samples do not come from paired mice. We did not have the resources to re-sequence using oxBS-seq or TAB-seq as well as our Nanopore sequence data. It was fortunate that publicly available data from homogenous tissue samples from inbred mouse lines were freely available. We assume a perfect sample set would have resulted in even lower rates of inter-assay deviation than we found. Action:

		 In order to avoid confusion, we have clearly stated that these samples are not the exact sample as those used in the public dataset (lines 195-197): “Samples were selected to match a publicly available archive of sequence data, containing whole genome oxidative bisulphite sequencing (oxBS-seq) for two biological replicates and TET-assisted bisulphite sequencing (TAB-seq) for three biological replicates (Ma et al., 2017).”
25	Regarding data availability, it would be nice that the authors make their POD5 data publicly available.	Reply: We agree with this suggestion and have made our murine sequence data publicly available via the NCBI Sequence Read Archive in fast5 format – as pod5 format is not yet accepted. Actions:  We have deposited these data on the NCBI Sequence Read Archive under the BioProject: PRJNA1144670. Additionally, the data is available in aligned BAM file format, which contains all modified base detections used in these analyses. Extracted modified bases in tabulated format are also now available on the NCBI GEO with accession GSE279860. We have referred to these publicly available data in the Code and data availability section (lines 690-694): “Raw nanopore sequence data in fast5 format, as well as aligned sequence data in BAM file format, has been made available for all murine samples on the NCBI Sequence Read Archive under BioProject PRJNA1144670. Additionally, CpG context modified base detections, as produced by ‘modkit pileup’ are available on the NCBI GEO archive with the accession: GSE279860. These data are limited to CpG positions relative to the mm39 reference genome and are soft-masked.”
26	Page 11 Line 337, is “(Table S2)” should be “(Table S3)”?	Reply: Thank you for catching this mistake. Action:  Supplementary table numbering has been corrected.

New section 1:

Accuracy of raw read detection of 5mC from methylation standards

Figure 1: Raw read accuracy of nanopore modified base detection using human whole genomic controls. a) Precision-Recall curve for 5mC detection using unmodified ($N=2$) and methylated controls ($N=2$). 5hmC detections are included as error, due to the absence of 5hmC in either control sample. Base-calls from both replicates of each control are counted and down sampled to 100,000,000 base-calls. b) False positive rate of modified base detection from the unmodified control as a function of local GC content. CpG base-calls are binned into non-overlapping 100bp windows and GC percentage calculated using the mm39 (GRCm39) reference genome. Bands indicate a 95% confidence interval across replicates ($N=2$). c-d) Logo representation of the 12-mer sequence up/down stream of false positive modified base detections for 5mC (c) and 5hmC (d). Higher base probabilities are shaded and stacked top-down. Includes reads from both unmodified DNA standards ($N=2$). e) Rates of false positive modified base detection across classes of repetitive, low-complexity, or CpG island sequences, including (top) 5mC and 5hmC false positives in the modification negative control ($N=2$), and (bottom) 5hmC false positives in the 5mC-positive control ($N=2$). LC: Low Complexity. SR: Simple Repeat. Error bars indicate 1 s.d. f) Confusion matrix for predicted and ground-truth base-calls. Base-calls across all replicates are counted, considering C to be the ground-truth state of all base-calls in the unmodified controls ($N=2$), and 5mC in the methylated controls ($N=2$). $**** p < 0.0001$

We sequenced two commercially available human DNA methylation standards (Zymo, D5013), a whole genome amplification (WGA)-produced modification negative control and an enzymatically methylated 5mC positive control. The modification negative control has no possibility of base modification; thus, all modified base detections represent false positives. The methylation positive control, which is enzymatically methylated after WGA, has a high degree of methylation ($> 95\%$) as reported by the manufacturer. All 5hmC detections represent classification errors. Raw read accuracy, which is the accuracy of modified base detection at a single base in one read, was very high for 5mC, with a precision of 0.99 and recall of 0.97 (Fig. 1a; Table 1).

Table 1: 5mC base detection statistics using methylation standards. Base-calls from both the unmodified negative standard and 5mC positive standard are concatenated and, for computational purposes, down-sampled to 100,000,000 base-calls.

True Positive Rate (TPR)	False Negative Rate (1-TPR)	False Positive Rate (FPR)	Precision $\left(\frac{TPR}{TPR+FPR}\right)$	Recall $\left(\frac{TPR}{TPR+FNR}\right)$	F1-score $\left(2 \left(\frac{Precision*Recall}{Precision+Recall}\right)\right)$
0.97	0.03	0.0056	0.99	0.97	0.98

Previous reports highlight a vulnerability of nanopore sequencing to mapping mismatches at regions of especially high GC content (Delahaye and Nicolas, 2021). To analyse its effect on modification detection, we inspected rates of false positive modification detection (FPR) as a function of local GC content. CpG detections were binned into 100bp windows, from which GC was calculated using the reference genome (hg38/GRCh38). This showed a strong correlation between GC content and FPR ($r = 0.76$; $p \geq 0.001$) (Fig. 1b), with GC contents substantially higher than the genomic mean (41.06%) experiencing the highest error rates.

Although we could not find any specific sequence motif in the 12-mer sequence up/downstream of a false positive modified base-call, there was a higher proportion of G or C base detections than expected from the genomic mean for both false positives of 5mC ($GC = 0.56$) (Fig. 1c) or 5hmC ($GC = 0.59$) (Fig. 1d).

Certain genomic elements may be predisposed to false positive error due to GC content; namely, Alu repeats (mean 51.3% GC) (Deininger, 2011), satellite repeats (mean 47.1% GC), and low complexity regions or simple repeat sequences (GC variable), such as tandem repeats. GC rich CpG islands (CGI) (mean 68.6% GC), which are important for transcriptional regulation, could also be vulnerable to this effect. Indeed, in the modification negative standard, FPR above the genomic mean was noted in low complexity repetitive sequences (Fig. 1e). By contrast, in the methylation positive standard (Fig. 1f), false 5hmC detection was largely consistent with the genomic mean 5hmC FPR for most contexts except for simple repeat elements, where 5hmC FPR appears to double.

False positive detection of 5hmC was present in both the unmodified and methylated standards. There was an almost five-fold difference in false 5hmC detection between the unmethylated (0.0024) and methylated (0.011) standards (Fig. 1g). This indicates a tendency for the base-caller to mistake true methylated positions for hydroxymethylation.

Actions:

- Added new figure to report accuracy statistics using modification negative and 5mC-positive DNA methylation standards (following Comment 3).

Figure and table revisions

Figure 2: CpG-resolution modified base detection using nanopore sequencing and bisulphite methods. Base-calls from matched CpG positions are concatenated across replicates in all plots. a-b) Density distributions of a) methylation, or b) hydroxymethylation, as the percentage modified reads at individual CpG sites, as found by nanopore sequencing (solid line) and the orthologous bisulphite method (dotted). Datasets are randomly down sampled to 1,000,000 CpG positions covered by each (merged) dataset for density computation. c) Histogram of deviations between matched CpG positions within the merged nanopore datasets and respective bisulphite orthologue for 5mC ($n=15,468,504$ CpG sites) or 5hmC ($n=11,283,580$). Deviations are expressed as the difference in the percentage of reads at any given base that are modified between techniques. Base-calls from matched CpG sites are merged across replicates. d) Violin plots showing CpG modification rate Z-score across different genomic contexts. Each violin contains a boxplot representing the distribution of modification Z-scores for that feature; for each, black box shows the interquartile range, white centre line indicates the median, and black lines extending to the 1.5x interquartile range. Base-calls from matched CpG sites are merged across replicates. Y-axis is limited between $-3 \leq Z \leq 3$ for visualisation. Table S3 displays the summary statistics of Z-score and percentage modification for each feature.

Actions:

- Figure number has been changed to reflect the addition of another Figure before this one.
- The empirical cumulative distribution plots (previously, a-b), have been changed to show density distribution plots (a-b) (Comment 8)
- Violin plots (d) have been created to replace the bar/line plots (previously, d-f) (Comment 16.)
- Figure legend has been changed to incorporate the above changes. More detail has been added to provide clarity for the violin plots.
- Random down-sampling was applied to the datasets to produce the density distribution plots shown in a-b. 10% of CpG sites were selected at random (respective to each dataset) to reduce computational load.
- Supplementary Table S3 has been added to display summary statistics of each context (regarding Comment 16).

Supplementary Figure 1: Benchmark dataset comparisons with sliding thresholds of minimum depth. Variation is calculated using the Root Mean Square Deviation (RMSD) between all possible dataset pairs, from which a mean is calculated to summarise all permutations. a) Variation in intra-assay variation as a function of depth. b) Change in mean CpG modification percentage over the same sliding depth threshold. c) Variation in inter-assay deviation between the nanopore datasets and respective bisulphite method for 5mC and 5hmC.

Actions:

- Added new supplementary figure related to the response to Comment 13 showing RMSD as a function of sampled sequencing depth for a) intra-assay error and c) inter-assay error. b) Notes the change in mean CpG methylation during as the sample depth threshold becomes higher, indicating an artefact of the oxBS and TAB-seq datasets.

Figure 1: Trial of direct nanopore sequencing of 5hmC-immunoprecipitated (IP) DNA. a) Representative PyGenomeTracks plot showing nanopore sequenced hMeDIP peaks overlapping the *Kcnj11* gene (chr7:45,746,000-45,751,000). For each repeat, tracks show the proportion of direct 5hmC calls at each CpG site (top) as well as sequencing coverage depth and the corresponding MACS2 narrow peak (bottom). Lowest tracks show WGS data from input. b) Bar plot of primary genomic context overlapped by peak regions, comparing the direct nanopore hMeDIP (N=3) with public hMeDIP-seq data (N=3). Genomic background, based on mm39, provided as reference. Error bars indicate 1 standard deviation. c) Bar plot showing modification states as a percentage of all CpG-context cytosine base-calls within the PromethION WGS data and hMeDIP peaks (N=3). Error bars indicate 1 standard deviation. d) Density plot underlay in shades of blue shows 5hmC Z-Score for matched 500bp windows of WGS data from nanopore and TAB-seq. Z-scores are calculated using the arcsine transformed proportion of all CpG base-calls (merged across replicates) enclosed in a window detected as 5hmC. Peaks from all nanopore hMeDIP-seq replicates are overlaid onto the window they intersect as a scatterplot, with size and hue proportional to fold enrichment over input. Dotted line indicates sample mean 5hmC Z-Score for the (x) TAB-seq and (y) nanopore datasets. * $p < 0.05$; ** $p < 0.01$; *** $p < 0.001$

Actions:

- Added dot plots to each bar in b) and c) to show replicate differences (per Figures and Tables requirements).
- After revising filtering thresholds to 5x (see response to Comment 13), the T-Test p-value for C bases as a proportion of all base-calls compared to WGS (3c), is now $p = 0.039381$. Accordingly, this bar is now marked with a star.

Figure 2: Duplex base-calling at CTCF-binding sites. a) CpG dyad duplex modification pattern across the whole genome, CTCF binding motifs, and ChIP-seq peak summits (split into two plot areas for visibility of dyad states with a small proportion of base-calls). Error bars indicate 1 s.d. Replicates are shown as separate dots (N=4). b) Violin plot representation of duplex modification pattern distance to CTCF ChIP-seq summit sites. CpG dyads more than 500bp from a summit are not represented. Duplex base-calls from all replicates are merged. c) Schematic representation of methylation and strand symmetry dependent CTCF loop formation. 1: A forward strand CTCF motif (left) overlaps a symmetrically unmodified CpG site and faces an asymmetrically methylated palindromic CTCF motif (centre). Methylation on the forward strand of the palindromic motif sequence favours CTCF binding in a reverse orientation, convergent with the previous unmodified motif and enabling loop formation. 2: Inversion of strand modification asymmetry at the palindromic motif favours CTCF binding in a forward orientation, becoming convergent with another unmodified CTCF motif (right). 3: Symmetrically unmodified CTCF motif enables binding in either or both orientations. ** $p < 0.01$; *** $p < 0.001$; **** $p < 0.0001$

Actions:

- Added dot plots to each bar in a) to show replicate differences (per Figures and Tables requirements).
- Figure legend changed to show replicates.

Supplementary Figure 2: Depth and breadth of coverage for whole mouse genome datasets. a-c) Depth of coverage at CpG positions (stranded) for each replicate in the a) nanopore sequence dataset, b) oxBS-seq dataset, or c) TAB-seq dataset. d) Comparison of median genomic sequencing depth depths across techniques, counting reads from either strand. e) Median depth at CpG positions, counting cytosine positions on opposing strands separately. f) Median CpG depth (stranded) as a function of local GC content. CpG sites were grouped into non-overlapping 100bp windows, from which GC percentage is calculated using the mm39 (GRCm39) reference genome. GC percentage is binned in 5-percentile intervals. The depth of matched CpG positions across replicates are plotted separately. Vertical line indicates genomic mean GC percentage of 41.7% in mouse. g) Percentage difference to the median CpG depth of a sequencing replicate within the bins defined in f). h) Percent difference between median CpG depth over different genomic contexts and median CpG depth of a sequencing replicate. i) Proportion of genome covered in at least 5x genomic depth (non-strand specific) by replicate.

Actions:

- Histograms have been replaced with line plots to indicate the depth of coverage in each sequencing replicate (a-c) (Comment 21).
- Plots have been added to present the overall differences in depth of coverage between sequence datasets (d-e).
- Plots have been added to explore the effects of GC distribution on depth of coverage between methods (f-g) as well as loss of coverage at regions typically affected by bisulphite sequencing (h) (Comment 7).
- A final plot was added to show overall coverage differences at a minimum threshold depth (i), with the nanopore dataset covering the most sites across replicates.

Supplementary Figure 3: Duplex modified base detection in nanopore sequence data. a) Count of all duplex and simplex-only reads across all Nanopore PromethION sequencing runs (N=8), using mouse cerebellum and all methylation standards. b-d) Pie charts show CpG dyad partners from the murine duplex datasets (CBM2_1, CBM2_2, CBM3_1, CBM3_2) as a percentage of all CpG dyad pairs containing a) unmodified C, b) 5mC, or c) 5hmC. Duplex base-calls across all replicates are counted.

Actions:

- Renamed murine replicates (CBM2_1, CBM2_2, CBM3_1, CBM3_2) for consistency with changes in SFig. 1.
- Legend changed to indicate replicates.

Supplementary Table 1: Depth and coverage statistics for mouse whole genome sequencing datasets. Genomic depth refers to depth of coverage in a non-strand-specific manner. Depth at CpG is the median depth of individual CpG-context cytosine positions on either strand.

	Biological replicate	Technical replicate	Median depth (genomic)	Median depth (CpG; stranded)
Nanopore PromethION	(1/2)	(1/2)	29	15
		(2/2)	33	17
	(2/2)	(1/2)	32	16
		(2/2)	29	16
TAB-seq	(1/3)	(1/1)	6	3
	(2/3)	(1/1)	8	3
	(3/3)	(1/1)	8	3
oxBS-seq	(1/2)	(1/1)	16	6
	(2/2)	(1/1)	5	2

Actions:

- Added new table of summary statistics for the median depth of each method (following Comment 4).

Supplementary Table 2: Summary statistics for modified base detection in all biological samples.

	Biological replicate	Technical replicate	Total CpG base-calls	Percentage detected as modified (%)	
				5mC	5hmC
Nanopore PromethION	CBM2	(1/2)	389,659,851	62.8	9.6
		(2/2)	442,519,412	62.6	9.6
	CBM3	(1/2)	446,149,544	60.3	10.0
		(2/2)	436,343,532	60.5	10.4
TAB-seq	(1/3)	(1/1)	71,651,039	NA	11.6
	(2/3)	(1/1)	77,410,573	NA	11.0
	(3/3)	(1/1)	86,804,432	NA	11.8
oxBS-seq	(1/2)	(1/1)	163,400,082	68.8	NA
	(2/2)	(1/1)	50,089,062	66.3	NA

Actions:

- Added new table of summary statistics for modification detection (related to Comment 9).

Table 2: 5mC base detection statistics using methylation standards. Base-calls from both the unmodified negative standard and 5mC positive standard are concatenated and, for computational purposes, down-sampled to 100,000,000 base-calls.

True Positive Rate (TPR)	False Negative Rate (1-TPR)	False Positive Rate (FPR)	Precision $\left(\frac{TPR}{TPR+FPR}\right)$	Recall $\left(\frac{TPR}{TPR+FNR}\right)$
0.97	0.03	0.0056	0.99	0.97

Actions:

- Table added. Related to Comment 3.

Table 2: Intra-assay variation of Nanopore sequencing (N=4), oxBS-seq (N=2), and TAB-seq (N=3). Root Mean Square Deviation (RMSD) is calculated pairwise between all possible replicate pairs. The mean of these pairwise comparisons is shown here for different depths.

		Intra-assay RMSD at x depth (%)			Mean count of sites compared across replicates		
		5x	10x	15x	5x	10x	15x
Nanopore	5mC	17.8	16.8	15.3	24,840,932	19,873,534	9,765,218
	5hmC	13.3	12.5	11.5	24,840,932	19,873,534	9,765,218
oxBS-seq		20.0	14.1	8.4	1,838,247	22,292	3,437
TAB-seq		16.3	11.3	6.3	2,813,280	105,536	6,126

Actions:

- Table added. Related to Comment 13.

Table 3: Inter-assay variation between replicates of Nanopore sequencing (N=4), oxBS-seq (N=2), and TAB-seq (N=3). Median Absolute Deviation (MAD) and Root Mean Square Deviation (RMSD) are calculated pairwise between all possible replicate pairs in both sequencing methods compared. The mean of these pairwise comparisons is shown here for different depths.

		Orthologous technique	MAD at x depth (%)			RMSD at x depth (%)		
			5x	10x	15x	5x	10x	15x
5mC	oxBS-seq		12.0	9.9	7.7	20.0	17.1	17.7
5hmC	TAB-seq		7.6	7.0	4.7	15.8	12.5	9.9

Actions:

- Table added. Related to Comment 13.

Supplementary Table 3: Summary statistics of 5mC and 5hmC base-calls in different genomic contexts.

	5mC				5hmC			
	Median (Z)	Mean (%)	Median (%)	Variance (%)	Median (Z)	Mean (%)	Median (%)	Variance (%)
3UTR	0.42	64.5	74.3	8.1	-0.21	11.6	8.5	1.2
5UTR	-1.49	7.8	0.0	4.5	-0.88	1.8	0.0	0.4
CGI	-1.77	6.7	0.0	4.1	-0.86	1.6	0.0	0.3
Exon	0.11	51.5	64.4	13.8	-0.49	8.7	4.9	1.2
Genic	0.08	59.2	70.5	10.5	-0.22	11.7	7.7	1.5
Intergenic	0.34	68.0	77.8	7.6	-0.44	8.2	5.1	0.9
Intron	0.33	61.2	71.6	9.4	-0.21	12.5	8.5	1.6
Promoter	-1.49	13.0	0.0	6.8	-0.88	3.3	0.0	0.6
Sea	0.36	68.4	77.3	6.9	-0.27	10.5	6.8	1.3
Shelf	0.06	59.7	68.8	8.9	-0.12	12.5	8.6	1.5
Shore	-1.35	32.0	17.6	11.4	-0.53	8.9	3.8	1.4

Actions:

- Table added in response to Comment 16.